EMBO
*reports*

# Reversible acetylation of HDAC8 regulates cell cycle

Chaowei Sang [ID] [1,2], Xuedong Li [ID] [1,2], Jingxuan Liu[1], Ziyin Chen [ID] [1], Minhui Xia [ID] [1], Miao Yu[1] & Wei Yu [ID] [1✉]

## Abstract

**HDAC8, a member of class I HDACs, plays a pivotal role in cell cycle regulation by deacetylating the cohesin subunit SMC3. While cyclins and CDKs are well-established cell cycle regulators, our knowledge of other regulators remains limited. Here we reveal the acetylation of K202 in HDAC8 as a key cell cycle regulator responsive to stress. K202 acetylation in HDAC8, primarily catalyzed by Tip60, restricts HDAC8 activity, leading to increased SMC3 acetylation and cell cycle arrest. Furthermore, cells expressing the mutant form of HDAC8 mimicking K202 acetylation display significant alterations in gene expression, potentially linked to changes in 3D genome structure, including enhanced chromatid loop interactions. K202 acetylation impairs cell cycle progression by disrupting the expression of cell cycle-related genes and sister chromatid cohesion, resulting in G2/M phase arrest. These findings indicate the reversible acetylation of HDAC8 as a cell cycle regulator, expanding our understanding of stress-responsive cell cycle dynamics.**

**Keywords** HDAC8; Acetylation; Cell Cycle; 3D Genome Structure; Sister Chromatid Cohesion
**Subject Categories** Cell Cycle; Chromatin, Transcription & Genomics; Post-translational Modifications & Proteolysis

## Introduction

The cell cycle represents a highly complex and tightly orchestrated process, intricately governed by a plethora of regulatory proteins. These proteins meticulously coordinate a precisely choreographed sequence of events that drive cellular division, culminating in the generation of two daughter cells (Gao et al, 2020; Li et al, 2021; Lieberman-Aiden et al, 2009). As is well-recognized, the cohesin complex assumes a pivotal role in the maintenance of chromosomal structural integrity and the faithful segregation of sister chromatids during the cell cycle. Beyond its role in cell division, the cohesin complex exerts regulatory control over its chromatin interactions in diverse cellular processes, encompassing DNA damage repair, gene transcription, and chromosome conformation (Horsfield et al, 2012). SMC3 functions as a core subunit of the cohesin complex, where precise regulation of its acetylation cycle is crucial for cohesin function. During the S-phase,

human SMC3 undergoes acetylation by ESCO1 and ESCO2, homologs of yeast Eco1. This acetylation process is reversible and is mediated by HDAC8 during prophase or anaphase, which is the human counterpart of yeast Hos1 (Beckouët et al, 2010; Ben-Shahar et al, 2008; Borges et al, 2010; Deardorff et al, 2012; Ünal et al, 2008; Xiong et al, 2010; Zhang et al, 2008). The deacetylation of SMC3 by HDAC8/Hos1 plays a critical role in recycling the cohesin complex for subsequent cell cycles (Beckouët et al, 2010; Ben-Shahar et al, 2008; Borges et al, 2010; Deardorff et al, 2012; Ünal et al, 2008; Xiong et al, 2010; Zhang et al, 2008). In addition, HDAC8 plays a pivotal role in shaping the 3D chromosome structure by regulating the size of architectural stripes and loops. Its deacetylation of SMC3 restarts the chromatin looping reaction by reducing the binding of PDS5A to SMC3 (van Ruiten et al, 2022). The deacetylation of SMC3 by HDAC8 is essential for various cohesin-related functions, and its diverse roles have been extensively documented (Kim et al, 2022). However, the precise mechanisms governing its enzymatic activity remain poorly understood.

Given that HDAC8 functions independently of additional cofactors (Hu et al, 2000), posttranslational modifications (PTMs) emerge as a promising regulatory mechanism governing HDAC8's functionality. PTMs play a pivotal role in modulating protease activity and orchestrating diverse cellular processes, offering the advantage of reversibility in response to fluctuations in the organism's energetic state or external environmental cues (Beltrao et al, 2012). Our understanding of the PTMs affecting HDAC8 has remained somewhat limited until recent times (Lee et al, 2004; Li et al, 2020). Acetylation, a prevailing PTM among proteins, merits profound investigation and scrutiny (Zhao et al, 2010). Nevertheless, the extent of acetylation modifications within HDAC8, especially concerning their regulatory implications during the cell cycle, remains an uncharted domain in current scientific inquiry.

In this study, we present evidence of reversible acetylation of HDAC8 as an intrinsic regulator of the cell cycle. Specifically, we highlight the acetylation of K202 in HDAC8 as a critical event that exerts a negative modulatory influence on its enzymatic activity. Notably, this acetylation event exhibits a strikingly dynamic pattern throughout the cell cycle. Interestingly, we also observe that external environmental stimuli have the capacity to promote Tip60-mediated K202 acetylation of HDAC8. This acetylation event, in turn, contributes to the accumulation of acetylation on SMC3, potentially exerting an influence on the progression of the cell cycle. Moreover, our findings demonstrate a compelling correlation between these regulatory events and changes in gene expression patterns, as well as alterations in the three-dimensional (3D) genome architecture. This interesting finding adds valuable insights to our understanding of the intricate mechanisms governing cell cycle regulation.

---

[1]State Key Laboratory of Genetic Engineering, School of Life Sciences, Zhongshan Hospital, Fudan University, 200438 Shanghai, China. [2]These authors contributed equally: Chaowei Sang, Xuedong Li. ✉E-mail: yuw@fudan.edu.cn

# Results

## K202 acetylation of HDAC8 inhibits its activity

In a previous investigation, we identified ten members of the HDAC family as targets for acetylation through high-throughput proteomic analysis, with HDAC8 being one of them. To gain insight into the functional acetylation sites that regulate HDAC8, we conducted mass spectrometry analysis using stable cells expressing Flag-tagged HDAC8. This analysis unveiled four potential acetylation lysine (K) residues, namely K52, K60, K202, and K370 (Fig. 1A,B). To assess the impact of acetylation at these sites on HDAC8's enzymatic activity, we expressed and purified recombinant human wild-type (WT) HDAC8 and the mutants K52Q, K60Q, K202Q, and K370Q, with 'Q' serving as a mimic for acetylation, all from *E. coli*. Strikingly, the mutants K60Q and K202Q displayed a pronounced reduction in enzymatic activity when compared to the WT HDAC8 (Fig. 1C). Notably, the K202

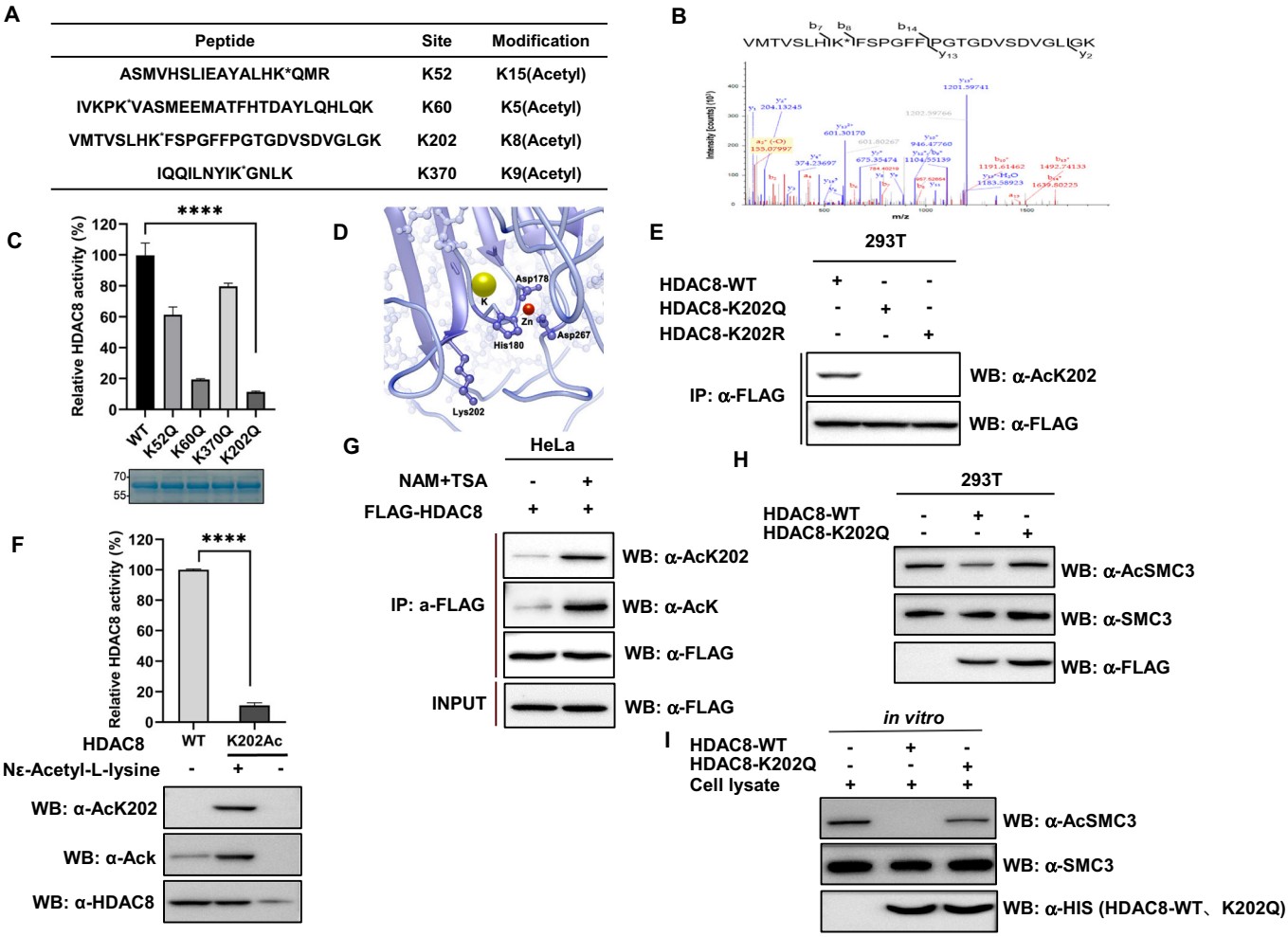

**Figure 1. K202 acetylation of HDAC8 inhibits its activity.**

(A) Identification of acetylated HDAC8 peptides by tandem mass spectrum. The specific acetylation sites are shown in the chart. (B) Acetylated K202 of HDAC8 was identified by tandem mass spectrum. The identified peptide is shown. (C) Acetylation of specific lysine sites in HDAC8 disrupts its deacetylase activity. Recombinant HDAC8 proteins were expressed in *E. coli*, purified using nickel affinity chromatography, and subjected to activity assay. Data were presented as mean ± SD ($n = 3$, technical replicates). Differences between groups were compared using unpaired two-tailed Student's *t* tests. ****$P < 0.0001$. (D) Localization of the K202 site in HDAC8 crystal structure (PDB accession code 1w22). The mutated residue was highlighted in red. (E) Representative immunoblotting of three independent experiments shows that HDAC8-K202Ac antibody specifically recognizes the K202 site acetylation of HDAC8. Acetylation levels of Flag-HDAC8 (WT), Flag-HDAC8 K202Q, and Flag-HDAC8-K202R expressed ectopically in 293T cells were measured using the site-specific HDAC8-K202Ac antibody. (F) The site-specific K202-acetylated HDAC8 exhibits a drastic decrease in deacetylase activity. Recombinant HDAC8-WT and K202Ac proteins were expressed in *E. coli*, purified using nickel affinity chromatography, and then detected by western blotting and performed activity assay. Data were presented as mean ± SD ($n = 3$, technical replicates). ****$P < 0.0001$. (G) Representative immunoblotting of three independent experiments shows that HDAC8 is acetylated. HeLa cells were transfected with Flag-tagged HDAC8 and then treated with 0.5 μM Trichostatin A (TSA) and 5 mM nicotinamide (NAM). Acetylation level of HDAC8 was detected using western blotting. (H) Representative immunoblotting of three independent experiments shows that HDAC8 K202Q mutant exhibits reduced deacetylation toward SMC3 in vivo. 293T cells were transfected with an empty vector, Flag-HDAC8 (WT), or Flag-HDAC8 K202Q. Results were detected by western blotting. (I) Representative immunoblotting of three independent experiments shows that HDAC8 K202Q mutant exhibits diminished deacetylation toward SMC3 in vitro. Whole-cell lysates were incubated with purified recombinant HDAC8-WT or K202Q mutant at 37 °C for 1h. Results were detected by western blotting. Source data are available online for this figure.

residue exhibits a high degree of evolutionary conservation, both across various species and within different class I HDACs (Fig. EV1A,B), hinting at a potentially pivotal role for K202 in HDAC8's functionality. Conversely, the conservation of the K60 residue was comparatively low (Fig. EV1C,D). Of particular significance, our simulated molecular structure analysis indicated that the K202 site resides in close proximity to the zinc-binding site crucial for catalytic activity (Fig. 1D). Subsequently, we proceeded to express and purify the K202R mutant of HDAC8 (where 'R' serves as a deacetylation mimic) from E. coli. Notably, the K202R mutant displayed a higher HDAC8 activity when compared to the K202Q mutant but still fell significantly short of the WT activity (Fig. EV1E). This observation underscores the critical role of the K202 site in modulating HDAC8 activity.

In accordance with previous research by Decroos et al, it has been established that K202 plays a pivotal role in establishing a crucial hydrogen bond network (D233-K202-S276) essential for HDAC8 activity (Decroos et al, 2015). By utilizing PyMOL to predict the structure of K202R/Q mutations, we observed that the K202R mutation disrupted the hydrogen bond with S276 and weakened the bond with D233, while the K202Q mutation led to the simultaneous loss of hydrogen bonds with both S276 and D233 (Fig. EV1F). These structural modifications may explain the differential residual activities observed for the K202R and K202Q mutants, approximately 58% and 12%, respectively (Fig. EV1E). Moreover, we conducted molecular docking assays between HDAC8 and its intracellular substrate, SMC3. In line with the current understanding of HDAC8's catalytic mechanism (Dowling et al, 2008; Somoza et al, 2004; Vannini et al, 2004), we found that the hydrophobic pocket of HDAC8 (WT) was occupied by the acetylated lysines (K105 and K106) of SMC3 during the deacetylation reaction. Intriguingly, both the K202R and K202Q mutations of HDAC8 interfered with the accessibility of SMC3's acetylated lysines to the pocket, as evidenced by the K105 and K106 sites of SMC3 being further away from the pocket of HDAC8 (Fig. EV1G). Notably, the interference caused by the K202Q mutation was more pronounced. Consistent with these observations, we noted that the K202R and K202Q mutants of HDAC8 exhibited progressively weaker binding to the substrate SMC3 in cells (Fig. EV1H). These findings suggest that K202R or K202Q mutations of HDAC8 may disrupt the structural stability by interfering with the hydrogen bond network, leading to weaker binding to substrates and consequently lower catalytic activity.

In order to effectively validate the K202 acetylation, we generated a specific antibody targeting acetylated K202 in HDAC8. Western blot analysis employing this site-specific antibody revealed a robust signal for ectopically expressed WT HDAC8 but no signal for the K202Q and K202R mutants in 293T cells (Fig. 1E). This confirmed the antibody's specificity in recognizing K202 acetylation. To provide unequivocal evidence of the impact of K202 acetylation on HDAC8 activity, we isolated recombinant HDAC8 proteins with complete acetylation at K202 in E. coli using a previously established expression system (Neumann et al, 2009; Wei et al, 2018). The enzymatic activity of HDAC8-K202Ac exhibited a substantial reduction of more than 90% compared to the WT HDAC8 (Fig. 1F). This reduction is consistent with the activity observed for the K202Q mutant, as depicted in Fig. 1C. In addition, we ectopically expressed Flag-tagged HDAC8 in HeLa cells and conducted immunoprecipitation. Western blotting with

the anti-HDAC8-K202Ac antibody and a pan-acetyl-lysine antibody confirmed that both pan-acetylation (Ack) and K202 acetylation (AcK202) levels were augmented by approximately five-fold following treatment with nicotinamide (NAM) and Trichostatin A (TSA) (Fig. 1G). To further affirm the impact of the K202Q mutation on HDAC8 activity in vivo, we overexpressed WT HDAC8 and the K202Q mutation in 293T cells. As anticipated, overexpression of the K202Q mutant did not induce significant changes in acetylated SMC3 levels, whereas overexpression of WT HDAC8 resulted in a roughly 50% reduction in acetylated SMC3 levels (Fig. 1H). Furthermore, by employing purified recombinant WT HDAC8 and the K202Q mutant, we assessed their capacity to deacetylate SMC3 in vitro. WT HDAC8 exhibited robust deacetylase activity toward SMC3, whereas the K202Q mutant demonstrated considerably diminished deacetylase activity (Fig. 1I). In sum, these findings substantiate that K202 acetylation functions to inhibit HDAC8 activity.

## Tip60-mediated K202 acetylation of HDAC8 is essential for acetylated SMC3 in the cell cycle

Numerous acetyltransferases have been implicated in the regulation of the cell cycle. In our quest to unravel the acetyltransferases responsible for catalyzing the acetylation of HDAC8, we initiated an examination of the interactions between HDAC8 and several nucleus-localized acetyltransferases, including PCAF, Tip60, and GCN5. Remarkably, our investigation revealed a robust interaction between Tip60 and HDAC8 (Fig. 2A,B). Subsequently, we directed our focus toward elucidating the role of Tip60 in the acetylation of HDAC8 at the cellular level. To accomplish this, we assessed the acetylation levels of ectopically expressed HDAC8 following transfection with either an empty vector or plasmids encoding Tip60 wild-type (WT) or Tip60-K327R, which is catalytically inactive (Probst et al, 2021). Notably, cells overexpressing Tip60 WT exhibited a significant increase in both pan-acetylation and K202 acetylation levels of HDAC8, concomitant with heightened acetylation of SMC3. Conversely, the Tip60-K327R mutant demonstrated weaker effects in enhancing pan-acetylation and K202 acetylation levels of HDAC8, along with SMC3 acetylation, compared to the overexpression of Tip60 WT (Fig. 2C). Moreover, when we employed siRNA to knock down Tip60, a notable reduction in the acetylation of HDAC8 and SMC3 was observed (Fig. 2D). Previous studies have established that the loss of HDAC8 activity disrupts the cell cycle (Dasgupta et al, 2016; Deardorff et al, 2012). In our endeavor to investigate the impact of K202 acetylation of HDAC8 on the cell cycle, we first established that the treatment of HeLa cells with various stressors, including $H_2O_2$, TSA, and glucose starvation, induced distinct degrees of G2/M phase arrest (Figs. 2E and EV2A). Subsequently, when HeLa cells ectopically expressed Flag-tagged HDAC8 were subjected to these stress conditions, this resulted in elevated levels of Tip60 and an increase in the acetylation of HDAC8 at K202 (Fig. 2F). Furthermore, ectopic overexpression of Tip60 (WT), but not the Tip60-K327R inactivating mutant, resulted in elevated acetylation levels of both HDAC8 and SMC3, accompanied by G2/M phase arrest. Collectively, these findings suggest that Tip60-mediated K202 acetylation of HDAC8 plays a crucial role in the acetylation of SMC3 and, consequently, in the regulation of the cell cycle.

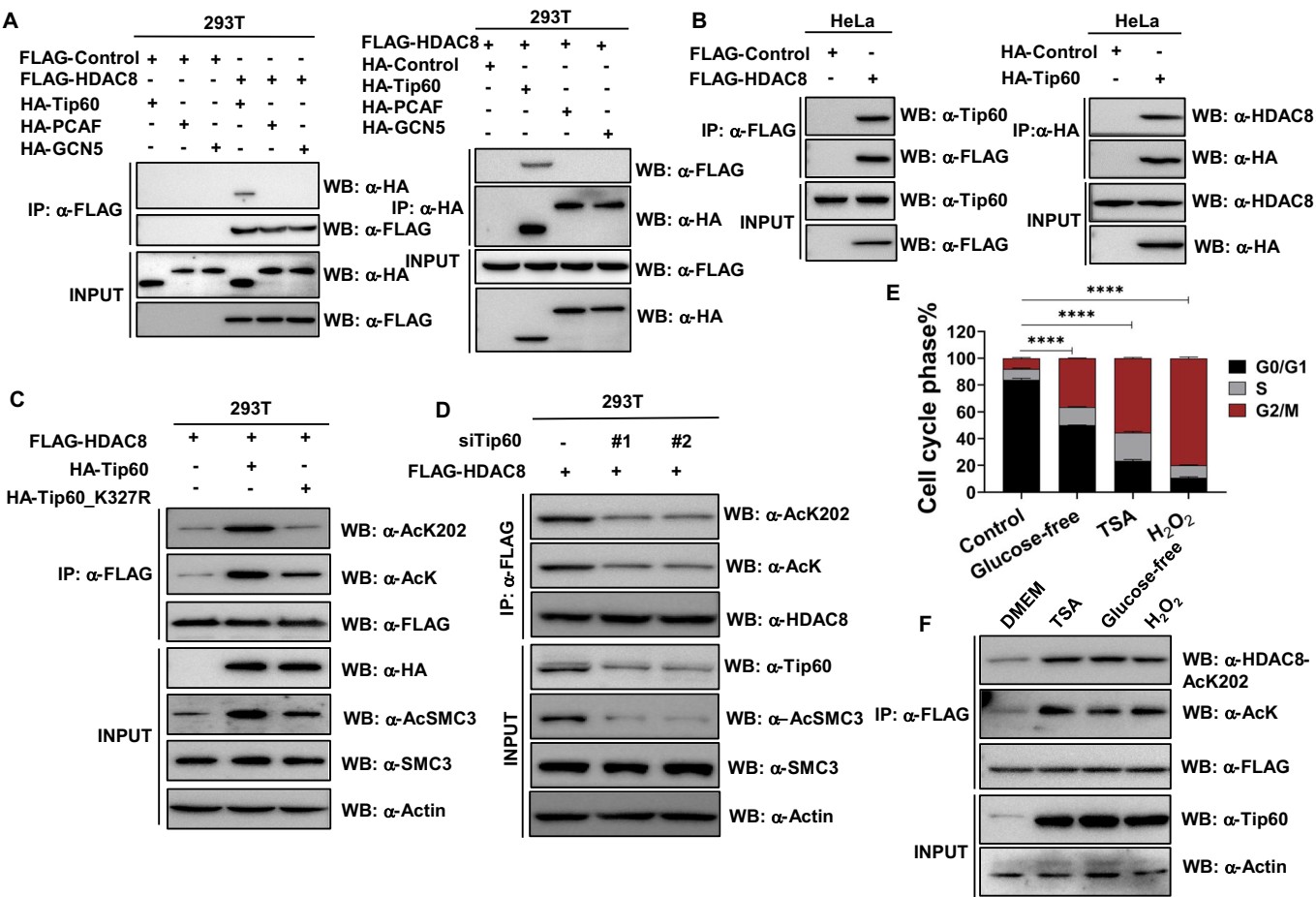

**Figure 2. Tip60-mediated K202 acetylation of HDAC8 is essential for acetylated SMC3 in the cell cycle.**

(A, B) Representative immunoblotting of three independent experiments shows that Tip60 interacts with HDAC8. The interaction between Tip60 and HDAC8 was investigated using western blot analysis in 293T cells (A) and HeLa cells (B). In 293T cells, the interactions were assessed by ectopically expressing Tip60 and HDAC8, while in HeLa cells, endogenous Tip60 and ectopically expressed HDAC8 were used. (C) Representative immunoblotting of three independent experiments shows that wild-type Tip60, not the inactive mutant Tip60 increases HDAC8 acetylation levels. In HeLa cells, Flag-tagged HDAC8 was ectopically expressed along with either wild-type Tip60, Tip60-K327R mutant, or an empty vector. Whole-cell lysates were subjected to immunoprecipitation with anti-Flag beads, followed by immunoblot analysis to detect the precipitated proteins. (D) Representative immunoblotting of three independent experiments shows that Tip60 knockdown decreases HDAC8 and SMC3 acetylation. 293T cells were transfected with Flag-HDAC8, and then acetylation levels of HDAC8 and SMC3 with or without Tip60 knockdown by siRNA were detected by western blotting. (E) Treatment with TSA, $H_2O_2$, or glucose-free medium induces cell cycle arrest at the G2/M phase in HeLa cells. The cells were initially synchronized in the early S-phase using a double-thymidine arrest and then exposed to TSA (500 nM), $H_2O_2$ (400 nM), glucose-free medium, or standard DMEM medium (as control) for 15 h. Flow cytometry analysis was performed with propidium iodide (PI) staining to detect cell cycle distribution. Results were presented as the mean ± SD ($n = 3$, biological replicates). Differences between groups were compared using unpaired two-tailed Student's $t$ tests. ****$P < 0.0001$. (F) Representative immunoblotting of three independent experiments shows that treatment with TSA, $H_2O_2$, or glucose-free medium induces Tip60 expression and HDAC8 acetylation. Asynchronous HeLa cells overexpressing Flag-tagged HDAC8 were treated with TSA (500 nM), $H_2O_2$ (400 nM), glucose-free medium, or standard DMEM medium (as control) for 15 h. Whole-cell lysates were subjected to immunoprecipitation with anti-Flag beads, followed by immunoblot analysis to detect the precipitated proteins. Source data are available online for this figure.

## K202 acetylation of HDAC8 is reversible

The acetylation modification process is known to be reversible. To further elucidate the characteristics of HDAC8-K202 acetylation, we subjected HeLa cells to different stressors, including TSA, $H_2O_2$, and glucose starvation, followed by subsequent western blotting and cell cycle distribution analysis. Under these various stress conditions, we observed a significant increase in HDAC8 acetylation levels, concomitant with elevated SMC3 acetylation levels, resulting in a pronounced G2/M phase cell cycle arrest (Figs. 3A,B and EV3A). Upon the removal of these stressors, the acetylation

levels of both HDAC8 and SMC3 reverted to their normal states, and the cell cycle resumed its typical progression, as demonstrated (Figs. 3A,C and EV3A). Similar alterations in the acetylation patterns of both HDAC8 and SMC3 were observed in cells ectopically expressed Flag-tagged HDAC8 with these stressors treatment or removal, and we also consistently observed concurrent changes in HDAC8 activity, strongly indicating that various stress stimuli could increase the acetylation of SMC3 by reducing HDAC8 activity (Figs. 3D and EV3B). Consequently, we sought to investigate whether Tip60 could mediate HDAC8 acetylation in response to stimulation. Previous studies have established that

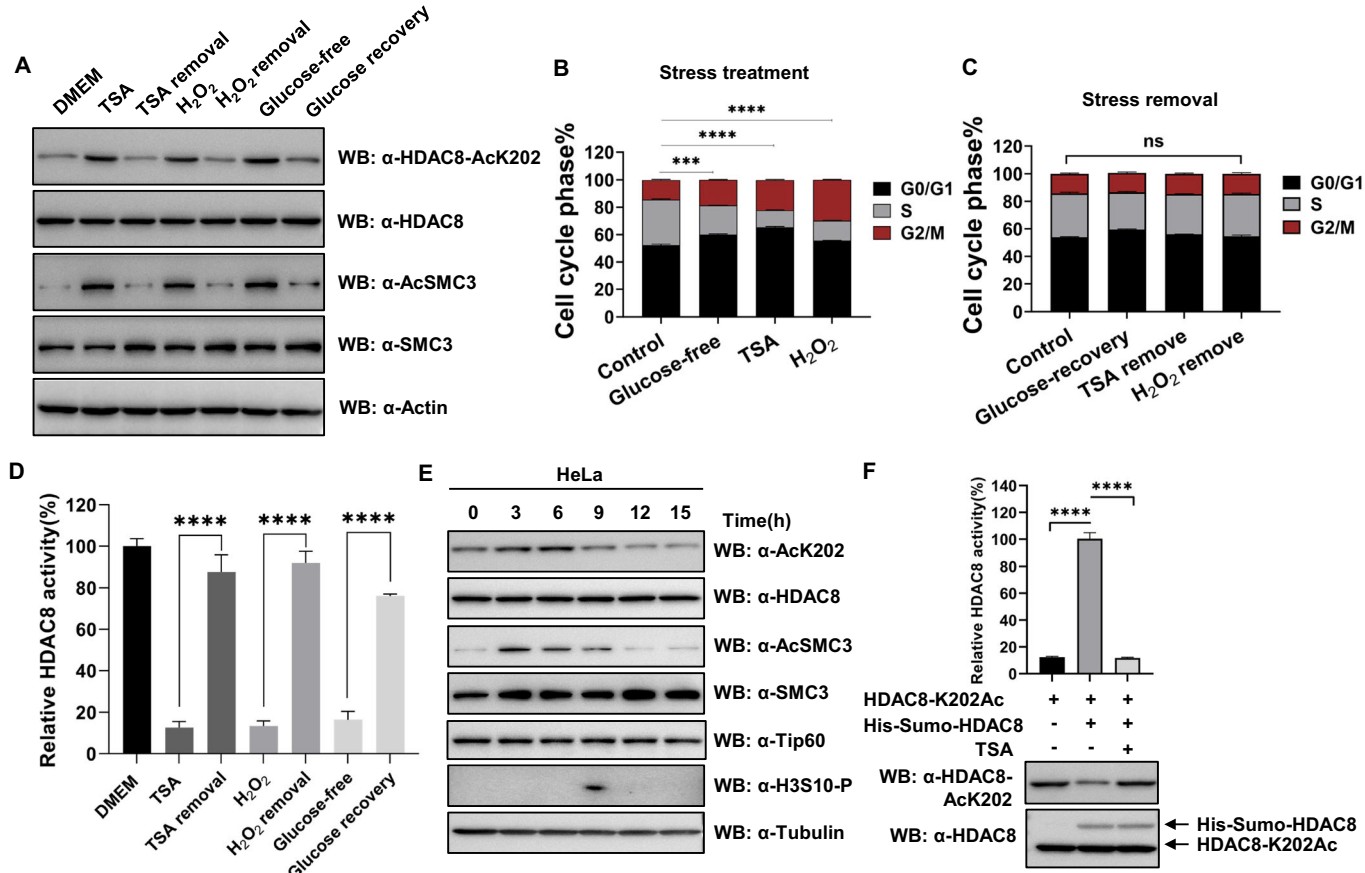

**Figure 3. K202 acetylation of HDAC8 is reversible.**

(A–C) Stress-driven K202 acetylation of HDAC8 is reversible. HeLa cells were treated with TSA (500 nM), $H_2O_2$ (400 nM), glucose-free medium, or standard DMEM medium (as control) for 24 h, and then after removing these stresses, the cells were re-cultured in the standard DMEM medium for about 48 h. Cells were harvested to prepare the whole-cell extracts for western blotting (A) or stain with PI for flow cytometry analysis (B, C) after stress treatment and removal. Results were presented as the mean ± SD ($n = 3$, biological replicates). Differences between groups were compared using unpaired two-tailed Student's $t$ tests. ***$P < 0.001$; ****$P < 0.0001$; ns, not significant. (D) Treatment with TSA, $H_2O_2$, or glucose-free medium caused a decrease of HDAC8 activity in HeLa cells. Asynchronous HeLa cells overexpressing Flag-tagged HDAC8 were treated with TSA (500 nM), $H_2O_2$ (400 nM), glucose-free medium, or standard DMEM medium (as control) for 24 h and then after removing these stresses, the cells were re-cultured in the standard DMEM medium for about 48 h. HDAC8 was enriched using Flag beads to perform activity assay. Results were presented as the mean ± SD ($n = 3$, technical replicates). Differences between groups were compared using unpaired two-tailed Student's $t$ tests. ****$P < 0.0001$. (E) Representative immunoblotting of three independent experiments shows that the acetylation level of HDAC8-K202 changes during the cell cycle progression. HeLa cells were synchronized in the early S-phase using a double-thymidine arrest, and the whole-cell extracts were prepared at different time points after release. Immunoblotting was performed to assess the levels of K202-acetylated HDAC8, total HDAC8, acetylated SMC3, total SMC3, Histone H3 Serine 10 phosphorylation (H3S10-P, a metaphase marker), and α-Tubulin. (F) HDAC8 deacetylates itself in vitro. Recombinant HDAC8-WT with a sumo tag and site-specific K202-acetylated HDAC8 (HDAC8-K202Ac) proteins were expressed in E. coil and purified using nickel affinity chromatography. HDAC8-K202Ac was incubated HDAC8-WT with or without TSA in the deacetylates buffer at 37 °C for 2 h, and then performed activity assay, and acetylation levels were detected by western blotting. Results were presented as the mean ± SD ($n = 3$, technical replicates). Differences between groups were compared using unpaired two-tailed Student's $t$ tests. ****$P < 0.0001$. Source data are available online for this figure.

glucose deprivation induces increased Tip60 expression in cells, as confirmed by both RT-qPCR and Western blot analysis (Hishikawa et al, 2019). In our study, we examined the endogenous expression of Tip60 in cells exposed to different glucose concentrations. Our findings were consistent with previous observations (Hishikawa et al, 2019). Our results supported an increase in Tip60 expression corresponding to decreasing glucose levels (Fig. EV3C), concomitant with elevated levels of SMC3 and HDAC8 acetylation. In addition, we examined acetylated HDAC8-K202 levels in glucose-starved cells with Tip60 knockdown. Remarkably, while Tip60 protein levels increased under glucose-free stimulation (Fig. EV3D, line 3 compared to line 1), the acetylated HDAC8-K202 returned to

normal levels under Tip60 knockdown conditions despite glucose starvation stimulation (Fig. EV3D, line 4 compared to line 2). These findings suggest that the stimuli triggering increased acetylation of HDAC8-K202 are dependent on Tip60. However, our further investigations revealed that knockdown of Tip60 in HeLa cells did not rescue cell cycle defects, indicating that Tip60 may fulfill multiple functional roles within the cell cycle (Fig. EV3E). In addition, we conducted a thorough examination of the K202 acetylation of HDAC8 during different phases of the cell cycle progression. In accordance with prior studies (Deardorff et al, 2012), the acetylation level of SMC3 exhibited fluctuations during the progression of the cell cycle. Surprisingly, we also noted similar

dynamic changes in the acetylation of HDAC8 at the K202 site (Fig. 3E). These observations imply that the acetylation and deacetylation of HDAC8, particularly at the K202 site, are subject to dynamic regulation. Furthermore, we found that the expression level of Tip60 remained constant throughout the cell cycle (Fig. 3E), suggesting that other regulatory mechanisms, such as posttranslational modifications, may govern Tip60 activity during different phases of the cell cycle. Previous research has demonstrated that HDAC8 can function as a deacetylase independently, without the need for additional cofactors (Hu et al, 2000). To investigate whether HDAC8 has the capability to deacetylate itself, we conducted co-incubation experiments involving HDAC8-K202Ac and HDAC8-WT protein tagged with SUMO in a deacetylation buffer. Through western blot analysis, we observed a reduction in HDAC8-K202 acetylation, accompanied by a notable increase in HDAC8 activity, which could be inhibited by TSA (Fig. 3F). Consequently, we conclude that the K202 acetylation of HDAC8 is indeed reversible, and HDAC8 possesses the capacity to deacetylate itself.

## K202 acetylation promotes chromatin loop formation that correlates with gene expression

To directly examine the influence of K202 acetylation on cellular processes, we harnessed CRISPR/Cas9 technology to modify the HDAC8 gene in HeLa cells. We generated the K202Q missense mutation, which mimics the acetylated state, and concurrently, we also generated the K202R mutant. Monoclonal cells were screened and confirmed for successful mutation of the HDAC8 gene through DNA sequence analysis (Fig. 4A). Notably, attempts to generate homozygote K202Q mutant cells were unsuccessful, thus heterozygous cells were used in our experiments. Conversely, we successfully obtained a homozygote K202R mutant. Whole-cell extracts from cells expressing wild-type (WT), K202R, and K202Q mutant forms of HDAC8 were prepared for western blot analysis. The results revealed that the K202R and K202Q mutations of HDAC8 led to progressively elevated levels of SMC3 acetylation, thereby confirming the successful introduction of mutations in HDAC8 in HeLa cells (Fig. 4B). These mutant cells were initially subjected to RNA-seq analysis, revealing a substantial number of differentially expressed genes between them (Fig. 4C). In particular, the K202Q mutant exhibited differential expression in 3439 genes, with 2437 genes upregulated and 1002 genes downregulated (Fig. EV4A). Recent research has provided insights into the critical role played by 3D genome structure in the regulation of gene activation and repression through various mechanisms, including the enhancer-promoter loops (Deng et al, 2022). To assess whether the K202R/Q mutants influenced 3D genome organization, two independent Hi-C experiments were performed using WT, K202R, and K202Q cells. We confirmed that harvested HeLa cells had a similar cell stage, ensuring that 3D chromosome structures were comparable in all conditions (Fig. EV4B). Subsequently, we comprehensively examined the chromatin interaction patterns between cells of three different genotypes at various levels. To gauge the contact probability between genomic regions at the chromosome level, we computed the relative contact probability (RCP), following the methodology (Lieberman-Aiden et al, 2009). Taking chromosome 1 as an example, the RCP curves across cells for three different genotypes showed a consistent overall trend,

where the interaction frequency decreased with increasing distance (Figs. 4D and EV4C). It is worth noting that the K202Q mutant cells exhibited more interactions at distances less than about 400 kb compared to the WT and K202R cells while fewer interactions at distances beyond about 400 kb. Given these observed differences, we applied aggregate TAD analyses (ATA) to investigate topologically associating domains (TADs) by aggregating Hi-C matrices around TADs. The K202Q mutant cells exhibited a slight increase in TADs compared to the WT and K202R cells (Figs. 4E and EV4D), which is in line with the higher frequency of interactions observed at distances less than about 400 kb. Notably, the K202Q mutant cells displayed a significant increase in contacts at the corners of TADs, which are often the sites of loop loci, as indicated by previous studies (Rao et al, 2014). Chromosome conformations are influenced by specific cell cycle dynamics (Nagano et al, 2017; Naumova et al, 2013), while recent research highlighting the general stability of chromatin loops during the G1 to S and G2 phases of the cell cycle (Nagano et al, 2017). Remarkably, our Hi-C results unveiled a substantial impact of HDAC8 K202Q mutations on chromatin loops. In comparison to the WT cells, the K202Q mutant cells exhibited a pronounced increase in loops (Figs. 4F and EV4E), align with recent observations made in HeLa cells treated with the HDAC8 inhibitor PCI (Nagasaka et al, 2023).

RCP analysis of two independent Hi-C experiments revealed consistent alterations in chromatin interactions in K202R mutant cells compared to WT cells, though these alterations were less pronounced than those observed in K202Q mutant cells. Notably, no evident changes were consistently observed in the aggregate intensity of TADs and chromatin loops between K202R and WT cells. Gene expression is intricately regulated, and the spatiotemporal aspects of genomic structure are increasingly recognized as pivotal for understanding eukaryotic gene expression. While the mechanistic underpinnings and causal links between structure and gene expression remain poorly understood, the development and application of Hi-C technology have provided valuable insights into such studies (Nollmann et al, 2022). In this study, we found that the normalized contact frequency of interactions showed a significant correlation with gene expression levels at distal elements of promoters. Upon comparing the promoter-centered interactions among the K202Q and K202R mutants and the WT cells, we observed that differences in the number of interactions for each promoter exhibited a positive correlation with the corresponding changes in gene expression (Figs. 4G and EV4F), as illustrated in individual maps (Fig. 4H) and corresponding gene expression levels (Fig. 4I). Importantly, this correlation was more pronounced between the K202Q mutants and the WT cells. In sum, both K202Q and K202R mutations have the capacity to influence chromatin interactions, while K202Q mutation exhibits a considerably more pronounced impact. Furthermore, the K202Q mutation prominently promotes the formation of chromatin loops, which are potentially associated with regulating the expression of related genes.

## K202 acetylation of HDAC8 impairs the cell cycle and causes cohesion defects

To gain deeper insights into how K202 acetylation of HDAC8 affects the cell cycle, further analysis of the RNA-seq data was conducted. Results revealed that the differentially expressed genes

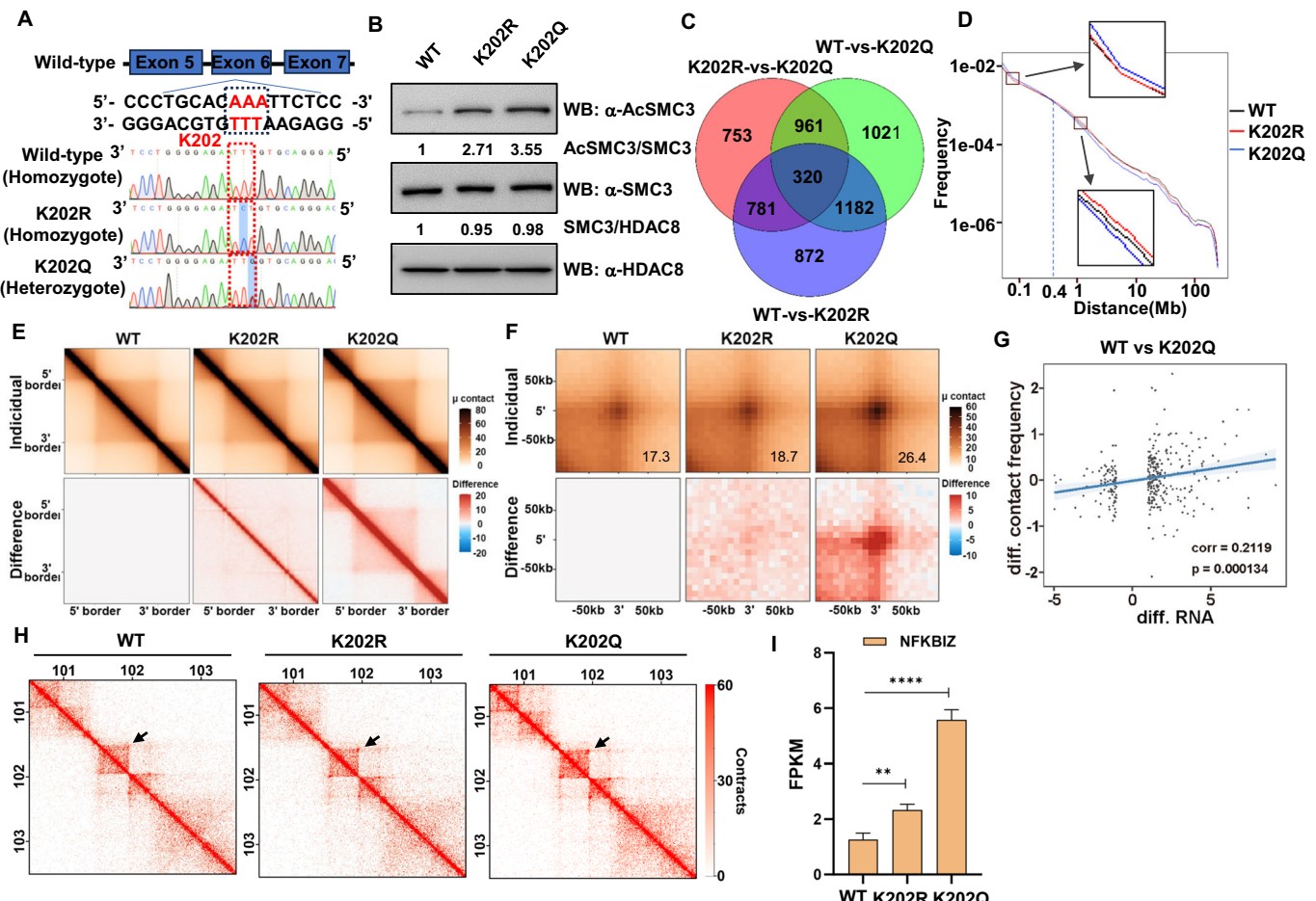

**Figure 4. K202 acetylation promotes chromatin loop formation that correlates with gene expression.**

(A) Site-specific mutations induced by the CRISPR/Cas9 in target gene HDAC8. Sequences of wild-type and representative mutation types induced at the K202 site, namely K202R and K202Q, are presented respectively. The red dotted box indicates the location of mutations. (B) Representative immunoblotting of three independent experiments shows that HDAC8 K202Q and K202R mutations cause elevated levels of SMC3 acetylation in the HeLa cell line. Whole-cell extracts of WT, K202R, and K202Q mutant cell lines were prepared, and the precipitated proteins were detected by western blotting. Numbers beneath bands indicate the quantification of normalized levels. (C) Volcano map of differentially expressed genes in the WT and K202Q cells. Significantly differentially expressed genes are shown as a red (up) or blue (down) dot. Non-significantly differentially expressed genes are shown as a gray dot. The horizontal and vertical coordinates represent the change of gene expression multiple and significance, respectively. (D) The relative contact probability (RCP) in chromosome 1 with a resolution of 40 kb for cells of the indicated genotypes. Representative areas are shown zoomed in. Valid contacts were normalized to 63 million among the three samples. (E) Aggregate TAD analyses (ATA) for TADs in the WT ($n = 4941$, median length 420 kb), K202R ($n = 4961$, median length 410 kb), and K202Q ($n = 5436$, median length 400 kb) cells. Genome-wide insulation score was computed and TADs were called at 10 kb resolution. Valid contacts were normalized to 63 million among the three samples. (F) Aggregate peak analysis (APA) for chromatin loops in cells of the indicated genotypes. Loops previously identified by Rao et al, (2014) ($n = 3094$, median length 220 kb) in wild-type HeLa cells were used to perform APA. Valid contacts were normalized to 63 million among the three samples. The bottom right value depicts the APA scores derived using the quantization function of the GENOVA to obtain the average signal strength of loops. (G) Scatterplot for the difference of the number of identified significant chromatin interactions (y axis) and the difference in gene expression (x axis, merged by log2(FPKM + 1)) between WT and K202Q cells. The blue solid line represents the fitted linear line, suggesting that the change of significant chromatin interactions positively correlates with the change in gene expression. (H) Hi-C contact matrices for cells of the indicated genotypes. A locus at chromosome 3 is shown at 10-kb resolution. The arrows indicate an example of the loop that has changed in K202R and K202Q compared with WT cells. Valid contacts were normalized to 63 million among the three samples. (I) Changes in the expression of the gene corresponding to the loop arrowed in (H) for cells of the indicated genotypes. Results were presented as the mean ± SD from RNA-seq data ($n = 3$, biological replicates). Differences between groups were compared using unpaired two-tailed Student's t tests. **$P < 0.01$; ****$P < 0.0001$. Source data are available online for this figure.

in both WT and K202Q mutant cells were significantly enriched in the PI3K-Akt signaling pathways, as indicated by KEGG enrichment analysis (Fig. 5A), which are related to the regulation of the cell cycle. Furthermore, our results showed that the expression of genes associated with the cell cycle was differentially altered in the K202Q and K202R mutant cells compared to WT cells. Notably, there was a noteworthy upregulation in the expression of genes

known to exert inhibitory effects on the cell cycle, such as p21 and p15. Conversely, genes associated with promoting cell cycle progression, including myc, exhibited a significant downregulation (Fig. 5B). We also performed gene set enrichment analysis (GSEA) of the differentially expressed genes, which revealed a significant enrichment of the cell cycle pathway (Figs. 5C and EV5A). These results indicated that K202R/K202Q mutations of HDAC8

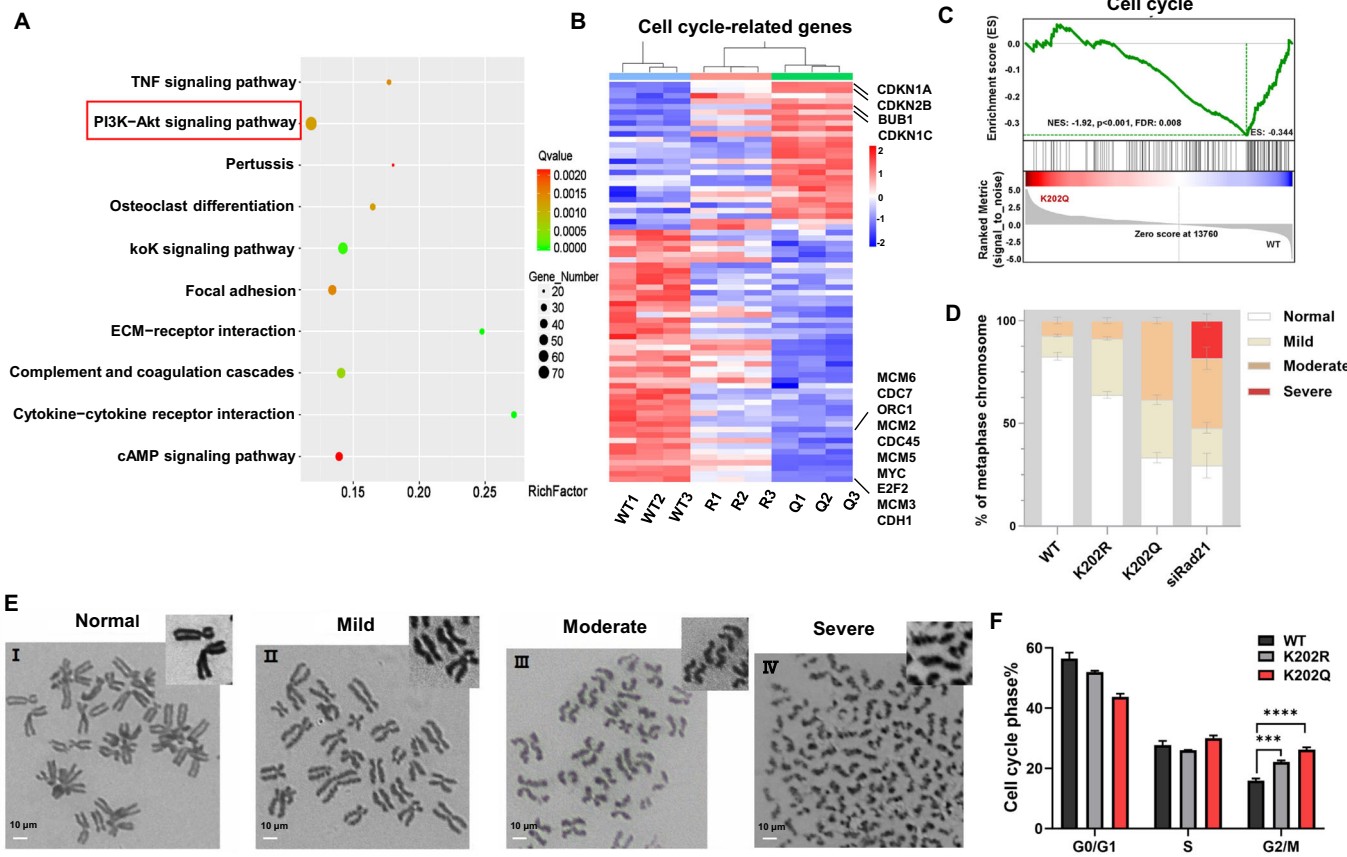

**Figure 5. K202 acetylation of HDAC8 impairs the cell cycle and causes cohesion defects.**

(A) Top ten enriched pathway terms are depicted in a bubble plot, where the y axis represents the pathway terms and the x axis represents the enrichment factor. The size and color of the bubble represent the number of differentially expressed genes enriched in pathway terms and enrichment significance, respectively. (B) Heatmap of RNA-seq data for cycle cell-related genes from WT, K202R mutation, and K202Q mutation cells (n = 3, biological replicates). Upregulated genes (greater than 1.2-fold) and downregulated genes (less than 0.83-fold) related to the cell cycle are represented. Color corresponds to the per-gene z-score calculated from log10 FPKM values, with red indicating upregulation and blue indicating downregulation. (C) Gene set enrichment analysis (GSEA) result of RNA-seq data in the HeLa cells of K202Q mutation with WT. The cell cycle signaling pathway is enriched with an FDR of 0.008. (D) K202Q cells exhibit pronounced cohesion defects. After being incubated with nocodazole overnight, the mitotic cells were collected by shake-off and analyzed by hypotonic spreading and Giemsa staining. The frequency of cohesion defects was assessed by analyzing 105–200 dividing cells in each experiment. Results were presented as the mean ± SD (n = 3, biological replicates). (E) Representative images of chromosome spread in HeLa cells. Cohesion was considered normal if sister chromatids were tightly connected at their centromeres. A majority of sister chromatids that showed abnormal spacing were considered cohesion defects, which were further categorized into mild, moderate, and severe. Individual chromosomes are shown at higher magnification in the right upper inset. Scale bars represent 10 μm. (F) K202Q cells exhibit G2/M phase arrest. HeLa cells of WT, K202R, and K202Q were cultured in the same conditions and then stained with PI for flow cytometry analysis. Results were presented as the mean ± SD (n = 3, biological replicates). Differences between groups were compared using unpaired two-tailed Student's t tests. ***P < 0.001; ****P < 0.0001. Source data are available online for this figure.

have clear negative regulatory effects on cell cycle progression. Importantly, the K202Q mutation displayed a stronger negative regulatory effect. To further investigate the impact on sister chromatid cohesion during the cell cycle, we performed the chromosome spread assay with siRad21 as a positive control (Fig. EV5B). We classified the observed cohesion phenotypes into four categories: normal, mild, moderate, and severe (Fig. 5E), and counted the frequencies of sister chromatid cohesion phenotypes (Fig. 5D). Knockdown of Rad21 using siRNA-induced cohesion defects, as described previously (Yang et al, 2019). K202Q cells displayed a notably high frequency of mild and moderate cohesion defects, accounting for ~28% and 39%, respectively. Remarkably, severe cohesion defects were rarely observed in K202Q mutation cells, contrasting with levels as high as 18% observed in cells subjected to Rad21 knockdown. An elevated frequency of cohesion

defects was also noted in K202R mutation cells, albeit predominantly characterized by mild cohesion defects and at a lower frequency compared to K202Q cells. Consistent with these findings, the proliferation rates of cells of different genotypes exhibited significant disparities. Acting as a positive control, siRNA-mediated knockdown of Rad21 notably suppressed cell proliferation, with optimal inhibition observed within 48 h, consistent with transient transfection dynamics. Relative to the WT cells, the proliferation rate of K202R cells exhibited a moderate decrease, whereas K202Q cells displayed a more pronounced inhibition of proliferation (Fig. EV5C). Furthermore, cell cycle analysis revealed that K202Q mutant cells exhibited a more pronounced G2/M phase arrest compared to K202R mutant cells and WT cells (Fig. 5F). Interestingly, these findings are in line with our observation that the K202Q mutation protein had a lower enzyme activity than

K202R (Fig. EV1E), suggesting that HDAC8 functions as a cell cycle regulator in an activity-dependent manner. In conclusion, the K202Q mutation severely impairs the cell cycle, as evidenced by the negative regulation of cell cycle-related genes and the heightened frequency of cohesion defects.

## Discussion

HDAC8 has emerged as a distinct deacetylase within the class I HDACs, exhibiting early divergence during evolution, which suggests a discrete functional specialization (Chakrabarti et al, 2015; Gregoretti et al, 2004; Wolfson et al, 2013). HDAC8 functions through the deacetylation of histones as well as a diverse range of non-histone proteins. Among its substrates, SMC3, a core subunit of the cohesin complex, was first identified in 2012 (Deardorff et al, 2012). During interphase mitosis, HDAC8-mediated deacetylation of SMC3 plays a crucial role in facilitating the reutilization of the cohesin complex during subsequent cell cycles. Here, we provide evidence supporting the reversible acetylation of HDAC8 as a novel cell cycle regulator. We have discovered that external stressors, such as glucose deprivation and hydrogen peroxide, trigger the K202 acetylation of HDAC8, which is mediated by Tip60. K202 acetylation inhibits HDAC8 activity, leading to elevated levels of SMC3 acetylation, which may cause cell cycle arrest. Importantly, our findings suggest that the cell cycle alterations are contingent on the acetylated state of HDAC8, which can be efficiently reversed when the external stressors are eliminated. These findings contribute to our understanding of cell cycle regulation and highlight the importance of HDAC8 in coordinating cellular responses to stress.

HDAC8 has been demonstrated to be involved in cell cycle regulation in both normal and cancer cells (Kim et al, 2022). Increased expression of HDAC8 is associated with cell proliferation in various cancer cells (Oehme et al, 2009; Song et al, 2015), whereas mutations in the HDAC8 gene have been linked to Cornelia de Lange syndrome (CdLS), which is a rare, genetically heterogeneous disorder that affects multiple organs and systems (Deardorff et al, 2012; Feng et al, 2014; Gao et al, 2018; Mio et al, 2021; Saikusa et al, 2018). Previous research has shown that PCI (a specific inhibitor of HDAC8 (Balasubramanian et al, 2008)) blocks the cell cycle progression in a concentration-dependent manner (Dasgupta et al, 2016), with minimal impact at low concentrations (Dasgupta et al, 2016; Deardorff et al, 2012; Ha et al, 2017). Interestingly, our study also revealed that HDAC8 appears to function as a cell cycle regulator in an activity-dependent manner. This can be supported by the observation that K202Q mutant cells exhibited more pronounced cell cycle disruptions than K202R mutant, compared to WT. Crucially, HDAC8-K202 acetylation exhibits dynamic alterations in a cell cycle-dependent manner, highlighting the pivotal role of HDAC8 activity regulation in governing cell cycle progression. Besides, most identified CdLS patients exhibit heterozygous mutations of the HDAC8 gene, implying that inactivation of key components of the cell cycle machinery is incompatible with cell proliferation. This observation may explain the unsuccessful attempts to generate homozygous K202Q mutant HeLa cells in our study. These findings imply that the K202Q mutation may be a novel missense mutation causing CdLS if such patients exist. Furthermore, our constructed cellular model may offer valuable insights for studying the pathogenic mechanisms of CdLS.

Gene expression is regulated by cohesin-mediated chromatin loops via long-range interactions between cohesin-binding sites and communications between enhancers and promoters (Furlongand Levine, 2018; Schoenfelderand Fraser, 2019; Zhuand Wang, 2019). Previous inquiries have unveiled the dynamic nature of chromatin structures across distinct cell cycle stages (Nagano et al, 2017; Naumova et al, 2013), while the loops predominantly manifest stability throughout the G1 to S and G2 phases (Nagano et al, 2017). In this study, we discerned a pronounced augmentation in the formation of these loops attributed to the K202Q mutation. Notably, our findings align with prior research indicating that the application of the HDAC8 inhibitor PCI augments the prevalence of loops in HeLa cells (Nagasaka et al, 2023) and that SMC3 acetylation serves to stabilize CTCF-anchored loops (Wutz et al, 2020). Remarkably, changes in gene expression were highly correlated with alterations in local chromatin interactions. Interestingly, K202 acetylation strengthened proximal interactions in the chromatin, while simultaneously attenuating distal interactions. Recent studies have suggested that HDAC8 catalyzes the deacetylation of cohesin subunit SMC3, promoting cohesin sliding across chromatin and thus favoring the formation of larger loops, i.e., more remote interactions (van Ruiten et al, 2022). This implies that the heightened K202 acetylation of HDAC8 impairs its ability to deacetylate SMC3, which consequently favors the stable binding of cohesin to DNA, restricting its sliding across chromatin. However, while consistent alterations in the frequency of chromatin interactions were observed in K202R mutant cells, alterations in the overall intensities of TADs and loops were not invariably apparent. This observation indicates that although the K202R mutation does alter the 3D structure of chromatin to some extent, its influence on the formation of TADs and chromatin loops is relatively limited. One plausible explanation is that the residual enzyme activity of the K202R mutation is largely sufficient to maintain overall TAD and loop regulation. Consequently, potential subtle deviations induced by the K202R mutation are challenging to detect reliably due to the inherent experimental noise associated with Hi-C techniques.

In sum, our investigation uncovers HDAC8 as an emerging cell cycle regulator. The activity of HDAC8 is negatively regulated by K202 acetylation, a reversible modification that undergoes regulation throughout the cell cycle. In response to external stress stimuli, an increased HDAC8-K202 acetylation level mediated by Tip60 causes elevated SMC3 acetylation, which may lead to cell cycle arrest. Furthermore, the regulation correlates with changes in gene expression and alterations in 3D genome structure. Our findings contribute to a deeper understanding of the regulatory mechanisms governing cell cycle progression.

## Methods

### Plasmid construction

Full-length cDNA of Human HDAC8 was amplified by PCR and cloned into indicated vectors including pRK7-Flag and pET28a-His-SUMO. Tip60, PCAF, and GCN5 were cloned into the pcDNA3.1-HA vector. Point mutations for HDAC8 and Tip60 were generated by site-directed mutagenesis (KOD Plus Mutagenesis Kit TOYOBO). All constructions were confirmed by DNA sequencing before further applications.

## Antibodies and reagents

Primary antibodies to Flag (Abcam 93713, 1:5000), HA (Abcam 9110, 1:5000), acetylated-lysine (Abcam 8227, 1:1000), Tip60 (Abcam 300521, 1:1000), Actin (Beyotime AA128, 1:5000), SMC3 (Abclonal A19591, 1:1000) and H3S10-P (Abclonal AP0840, 1:1000) were commercially obtained. Antibody against AcSMC3 is obtained from previously described in (Deardorff et al, 2012). Antibodies specific to HDAC8 and HDAC8 K202Ac were prepared commercially from different immunizing rabbits at GL Biochem (Shanghai) Ltd. (HDAC8: antigen peptide: CHKFSPGFFPG TGDVSD, 1:1000; HDAC8 K202Ac: antigen peptide: CHK(ac) FSPGFFPGTGDVSD, 1:1000). NAM (Sigma N0636), TSA (Sigma V900931), PI (Sigma P4170) and N-acetyl-lysine (Sigma A4021) are commercially obtained.

## Cell culture and transfection

HeLa and 293T cell lines were from the Fudan Molecular and Cell Biology (MCB) research laboratory. Cells were cultured in DMEM (Gibco) supplemented with 10% FBS (BI), 100 units ml$^{-1}$ penicillin, and 100 µg ml$^{-1}$ streptomycin (Sangon Biotech). Cells were maintained in an incubator at 37 °C, in a humidified atmosphere containing 5% CO$_2$. Cell transfection except for siRNA was performed using polyethylenimine (PEI) (Polysciences) or Lipo-Max (Sudgen) based on a 3:1 ratio of PEI/LipoMax (µg): total DNA (µg), according to the manufacturer's protocol. Cell transfection for siRNA was carried out by Lipofectamine RNAiMAX (Thermo) according to the manufacturer's protocol. Synthetic siRNA oligonucleotides were as follows: siRNA_Tip60#1: 5'-CCTCCTATC CTATCGAAGCTA-3'; siRNA_Tip60#2: 5'-CCTCAATCTCATC AACTACTA-3'; siRNA_Rad21: 5'-GGCCAGCAGAACAAAC AUAUU-3'.

## Synchronization of HeLa cells

HeLa cells were synchronized in the early S-phase by double-thymidine arrest. Briefly, HeLa cells of 20–25% density were cultured with 2 mM thymidine medium for 18 h, washed twice with PBS, and cultured in standard medium for 9 h. For the second arrest, cells were cultured with 2 mM thymidine medium for another 18 h. After release from the arrest, cells were harvested for western blotting at the indicated time points.

## Flow cytometric analysis

Cells were treated with 70% alcohol at 4 °C overnight. After washing twice with pre-cooled PBS, cells were then incubated with propidium iodide (PI) solution (containing 20 µg/mL PI, 20 µg/mL RNAase, and 0.1% Triton X-100, dissolved in PBS) for about 30 min at room temperature. Cells were analyzed using a fluorescence-activated cell sorter (FACS) (BD FACSCalibur).

## Targeted genome editing

HDAC8-K202R/K202Q mutation cell lines were established using the CRISPR/Cas9 system. sgRNA sequences (5'-GGACACGGT CATGACTTTGG-3') targeting HDAC8 were designed by a CRISPR designer at http://crispr.mit.edu/, and cloned into the CRISPR/Cas9 plasmid vector PX330-mCherry. A 1.5 kb 5' homology arm and a 2 kb 3' homology arm containing a specific mutation in the exon 6 of HDAC8, were cloned into the pEGFP-N1 vector used for homologous directed DNA repair. Both constructed plasmids were transfected together into HeLa cells. Cells were sorted with FACS (SONY-MA900), and single colonies were picked after 10–14 days of incubation and validated by DNA sequencing.

## Immunoprecipitation, co-immunoprecipitation, and western blot

For cell-based experiments, cells were washed twice with pre-cooled PBS and then lysed with 1% Nonidet P40 buffer containing 50 mM Tris-HCl (pH 7.5), 150 mM NaCl, and multiple protease inhibitors (PMSF 1 mM, Aprotinin 1 µg ml$^{-1}$, Leupeptin 1 µg ml$^{-1}$, Pepstatin 1 µg ml$^{-1}$, Na$_3$VO$_4$ 1 mM, NaF 1 mM, and additional TSA 2.5 µM and NAM 25 mM were needed for experiments to detect acetylation). For immunoprecipitation or co-immunoprecipitation, whole-cell lysates were incubated for 4 h at 4 °C with anti-Flag/HA beads after removing debris by centrifuging at 4 °C, 13,000 rpm for 20 min. The beads containing immunoprecipitates underwent three washes with lysis buffer, with centrifugation at 200 rpm for 2 min after each wash. Then beads were boiled with 1×SDS loading buffer, loaded on SDS/PAGE, and transferred onto nitrocellulose membrane (GE Healthcare 10600002) for western blot analysis.

## Recombinant human HDAC8 expression and purification

The cDNA encoding human HDAC8 (isoform 1; NCBI Reference Sequence: XP_006711987.1) was cloned into the pET28a vector with an additional SUMO tag after His tag in N-terminal. Recombinant HDAC8 was expressed in *E. coli* strain BL21 (DE3) and purified according to published procedures (Deardorff et al, 2012), with minor modifications. In brief, cells were grown in LB-medium containing 50 µg/mL kanamycin at 37 °C overnight and then were inoculated as 1:100 into 1-L flasks. Cultures were grown at 37 °C with shaking until OD$_{600}$ ~0.5, at which point the cells were induced by the addition of isopropyl β-D-thiogalactopyranoside (0.4 mM final concentration) and ZnCl$_2$ (100 µM final concentration), and grown overnight at 18 °C. Cells were harvested by centrifugation, washed with PBS, and then kept at −80 °C until purification. After thawing, cells were resuspended in 30 ml lysis buffer (50 mM Tris-HCl pH 8.0, 500 mM KCl, 5% glycerol, 3 mM 2-ME, and 100 µM phenylmethanesulphonyl fluoride). Cells were then lysed through high-pressure disruption by a low-temperature Ultra-High-Pressure Continuous Flow Cell Disrupter (JN-3000 plus), and the cell lysate was centrifuged at 18,000 rpm for 1 h at 4 °C. The supernatant was loaded onto a 5 ml Ni-Sepharose column His Trap-HP (GE Healthcare) by a peristaltic pump and gradient eluted by an ÄKTA FPLC system (GE Healthcare) with elution buffer (Buffer A: 50 mM Tris-HCl pH 8.0, 500 mM KCl, 5% glycerol, 3 mM 2-ME and 100 µM phenylmethanesulphonyl fluoride; Buffer B: 50 mM Tris-HCl pH 8.0, 500 mM KCl, 5% glycerol, 3 mM 2-ME and 100 µM phenylmethanesulphonyl fluoride and 250 mM imidazole). To discard the SUMO tag, collected eluates were subjected to ULP enzyme digestion at 4 °C overnight. Expression and purification of recombinant HDAC8-K202R/Q mutant were conducted as described above.

## Genetically encoding Nε-acetyl-lysine in recombinant proteins

To generate a homogenously K202-acetylated HDAC8 construct, we used a three-plasmid (TEV-8, pCDFpylT-1, and pAcKRS) system as previously described (Neumann et al, 2009; Wei et al, 2018). Briefly, we cloned wild-type HDAC8 into pTEV-8 producing a C-terminal His6-tagged construct, and incorporated an amber codon at lysine 202 (AAA to TAG by site-directed mutagenesis). The amber construct was overexpressed in LB-medium with spectinomycin (50 mg ml$^{-1}$), kanamycin (50 mg ml$^{-1}$), and ampicillin (150 mg ml$^{-1}$), in addition to 2mM N-acetyl-lysine and 20 mM NAM at the time of induction. Cell culture, expression, and purification were referred to as recombinant human HDAC8 methods as described above.

## HDAC8 activity assay

HDAC8 activity was measured using the commercially available fluorescent deacetylase substrate Ac-Lys-AMC (Enzo). A typical assay mixture in a 96-well plate containing 2 mM Ac-Lys-AMC and the enzyme in the deacetylase buffer (50 mM Tris pH 8.0, 4 mM MgCl$_2$, and 0.2 mM DTT) was incubated at 37 °C. After 2 h, the reactions were quenched by the addition of the Trypsin (Sigma, 2.5 mg ml$^{-1}$ final concentration), followed by further incubation for 1 h at 37 °C. Fluorescence was measured with excitation at 360 nm and emission at 460 nm, using either the Synergy2 or Cytation3 microplate reader (BioTek).

## Quantitative real-time PCR

Total RNA was extracted using TRIZOL (Life Technologies, 15596018CN) and reverse-transcribed into cDNA using the qPCR RT Master Mix with gDNA Remover kit (TOYOBO). cDNA was then used as a template for real-time PCR, performed with SYBR Green Real-time PCR Master Mix (TOYOBO) on a CFX Connect Real-Time System (Bio-Rad). mRNA levels were quantified using the ΔΔCt method and normalized to GAPDH as an internal control. Primer sequences for each gene are provided below: RAD21 (forward): 5′-TCATGGTCTTCAGCGTGCTC-3′; RAD21 (reverse): 5′-ACTTTGCGGCAGCTTGTTTT-3′; GAPDH (forward): 5′-AAGCTCATTTCCTGGTATGACAA-3′; GAPDA (reverse): 5′-CTTACTCCTTGGAGGCCATGT-3′.

## Chromosome spreads

Chromosome analysis was prepared as previously described (Deardorff et al, 2012), with minor adjustments. In brief, HeLa cells were treated with nocodazole (100 ng/ml) overnight. Mitotic cells were collected by mitotic shake-off and then hypotonically swollen in 40% PBS and 60% tap water for 5 min at room temperature. Subsequently, cells were fixed with freshly made Carnot's solution (methanol: acetic acid = 3:1) and dropped onto glass microscopy slides for natural drying. Slides were stained with 5% Giemsa (Yuanye R20653) solution for 10 min, washed with tap water, air-dried, and mounted using Mounting Medium (Yeasen 36313ES60). Images were captured using an ECLIPSE Ni-U microscope equipped with 100× oil objective.

## RNA-Seq and data analysis

In total, 1 μg total RNA was extracted using TRIZOL (Life Technologies 15596018CN) from HeLa cells ($n = 3$). Libraries were generated from the total RNA using the VAHTS Universal V8 RNA-seq Library Prep Kit for Illumina (VAHTS NR605). Sequencing was carried out on the Novaseq 6000 platform using a 2 ×150 bp paired-end (PE) configuration. Sequenced reads were aligned to the hg38 Homo sapiens reference via software Hisat2 (v.2.0.1). Differential expression analysis was performed using the DESeq2 (v.1.26.0). Gene Set Enrichment Analysis (GSEA) was performed using GSEA software (v.4.3.2). Enrichment analysis of KEGG signaling pathways was conducted by using the Database for Annotation, Visualization, and Integrated Discovery (DAVID, http://en.wikipedia.org/wiki/KEGG).

## Hi-C library

Hi-C libraries were prepared using in situ Hi-C protocol as previously described (Rao et al, 2014) with some modifications. In brief, 50–500 thousand HeLa cells were crosslinked with 16% formaldehyde (Thermo 28908) to a final concentration of 1% for 10 min at room temperature, followed by quenched by 0.2 M glycine for 5 min. Cell nuclei were extracted with lysis buffer (10 mM Tris-HCl pH 8.0, 10 mM NaCl, 0.2% Igepal CA630) mixed with 50× protease inhibitor (Sigma P8340) on ice for at least 15 min. Cell pellets were resuspended in 0.5% SDS (prepared from 10% SDS (Sigma 71736)) and incubated at 62 °C for exactly 10 min, then quenched with 1.5% trix-100 (freshly prepared from 10% Triton X-100 (Sigma 648463)) for 15 min at 37 °C. DNA was then digested with Dpn II restriction enzyme (NEB R0543L) for 16 h at 37 °C, followed by an incubation at 62 °C for 20 min. End repair of DNA as well as biotin labeling was achieved by incubation at 37 °C for 1 h in the fill-in master mix (with a final concentration of 0.4 mM biotin-14-dATP (Invitrogen 19524016), 3.3 mM dCTP, 3.3 mM dGTP, 3.3 mM dTTP, 5000 U/mL DNA Polymerase I, Large (Klenow) Fragment (NEB M0210L)). DNA was then ligated by incubation in a ligation master mix (composed of 664ul of water, 120 μl of 10X T4 DNA ligase buffer (NEB B0202S), 100 μl of 10% Triton X-100, 6 μl of 20 mg/ml Bovine Serum Albumin (Sigma B8667), and 10 μl of 400 U/μl T4 DNA Ligase (NEB M0202S)) at room temperature with low-speed rotation for 4 h. Cell pellets were resuspended in 300 μL of PK digestion buffer (10 mM Tris-HCl pH 8.0, 0.5 M NaCl, 1% SDS), followed by the addition of 50 μL of 20 mg/ml proteinase K (QIAGEN 19133). The mixture was digested at 55 °C for 30 min to uncross-link the DNA, and then incubated overnight at 68 °C. To efficiently purify the DNA, the TIANamp Genomic DNA Kit (TIANGEN DP304) was used instead of low-temperature precipitation with ethanol and sodium acetate. Sonication was then employed to break DNA into an average size of 200–400 bp. Biotin-labeled DNA was pulled down with Dynabeads MyOne Streptavidin C1 beads (Invitrogen 65001) and prepped for library construction. Libraries were constructed using the VAHTS Universal DNA Library Prep Kit for Illumina V3 (Vazyme ND607) following the instructions. Libraries were sequenced as 150 bp paired-end reads on NovaSeq 6000 sequencing platforms.

## Hi-C analysis

Data was analyzed similarly to previously (van Ruiten et al, 2022). In brief, Hi-C-Pro v.3.1 (ref. (Servant et al, 2015)) was used for data

preprocessing with hg19 Homo sapiens as a reference. Reads were processed by Juicebox-pre (juicer tools v.1.9.9) (Durand et al, 2016) to generate Juicebox-ready files. Contact matrices were ICE normalized for downstream analyses. By GENOVA v.1.0 (van der Weide et al, 2021), TADs were called at 10 kb resolution with "w" as 25, and then ATA analysis was performed. APA(Rao et al, 2014) was also performed in GENOVA v.1.0, using the loops previously identified in wild-type HeLa cells (Rao et al, 2014). The Relative Contact Probability (RCP) (Lieberman-Aiden et al, 2009) was computed at 40 kb resolution by the GENOVA package.

To analyze the correlation between changes in gene expression and chromatin interactions, we initially selected differentially expressed genes (DEGs) with more than twofold changes between two samples. We defined the ±1 kb regions around the transcription start site (TSS) of genes as the approximate promoter regions and overlapped the promoters of DEGs with the loops previously identified in wild-type HeLa cells (Rao et al, 2014) under 25 kb resolution. We extracted the normalized contact frequencies of these loops in our samples. and calculated the Pearson correlation coefficient (PCC) and the corresponding $p$-value between the changes in gene expression and the changes in normalized contact frequency in chromatin loops.

### Statistical analysis

Data were analyzed using GraphPad Prism 10.0 or R statistical program as indicated in the figure legends. Results were presented as the mean ± standard deviation (SD) of three duplicate experiments. Differences between groups were compared using unpaired two-tailed Student's $t$ tests. Significance levels were denoted as follows: $*P < 0.05$; $**P < 0.01$; $***P < 0.001$; $****P < 0.0001$; ns, not significant for the indicated comparison.

## Data availability

Both Hi-C and RNA-seq data generated have been deposited in GEO, GEO accession: GSE247611 (https://www.ncbi.nlm.nih.gov/geo/query/acc.cgi?acc=GSM7897979).

The source data of this paper are collected in the following database record: biostudies:S-SCDT-10_1038-S44319-024-00210-w.

## Peer review information

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

## Acknowledgements

The authors thank members of Yu Lab for their technical support throughout this study. We thank the facility center of the State Key Laboratory of Genetic Engineering for technical support, specifically the flow cytometry core facility. We thank Prof. Lin Jinzhong for the high-throughput computing and storage assistance. This work was supported by the National Key R&D Program of China (2023YFA1800400) and the National Natural Science Foundation of China (32370825, 92249302, 92049301, 31821002).

## Author contributions

**Chaowei Sang**: Data curation; Investigation; Writing—original draft; Writing—review and editing. **Xuedong Li**: Data curation; Investigation; Writing—original draft. **Jingxuan Liu**: Data curation. **Ziyin Chen**: Data curation. **Minhui Xia**: Data curation. **Miao Yu**: Data curation; Methodology. **Wei Yu**: Conceptualization; Resources; Formal analysis; Supervision; Funding acquisition; Writing—original draft; Project administration; Writing—review and editing.

Source data underlying figure panels in this paper may have individual authorship assigned. Where available, figure panel/source data authorship is listed in the following database record: biostudies:S-SCDT-10_1038-S44319-024-00210-w.

## Disclosure and competing interests statement

The authors declare no competing interests.

# Expanded View Figures

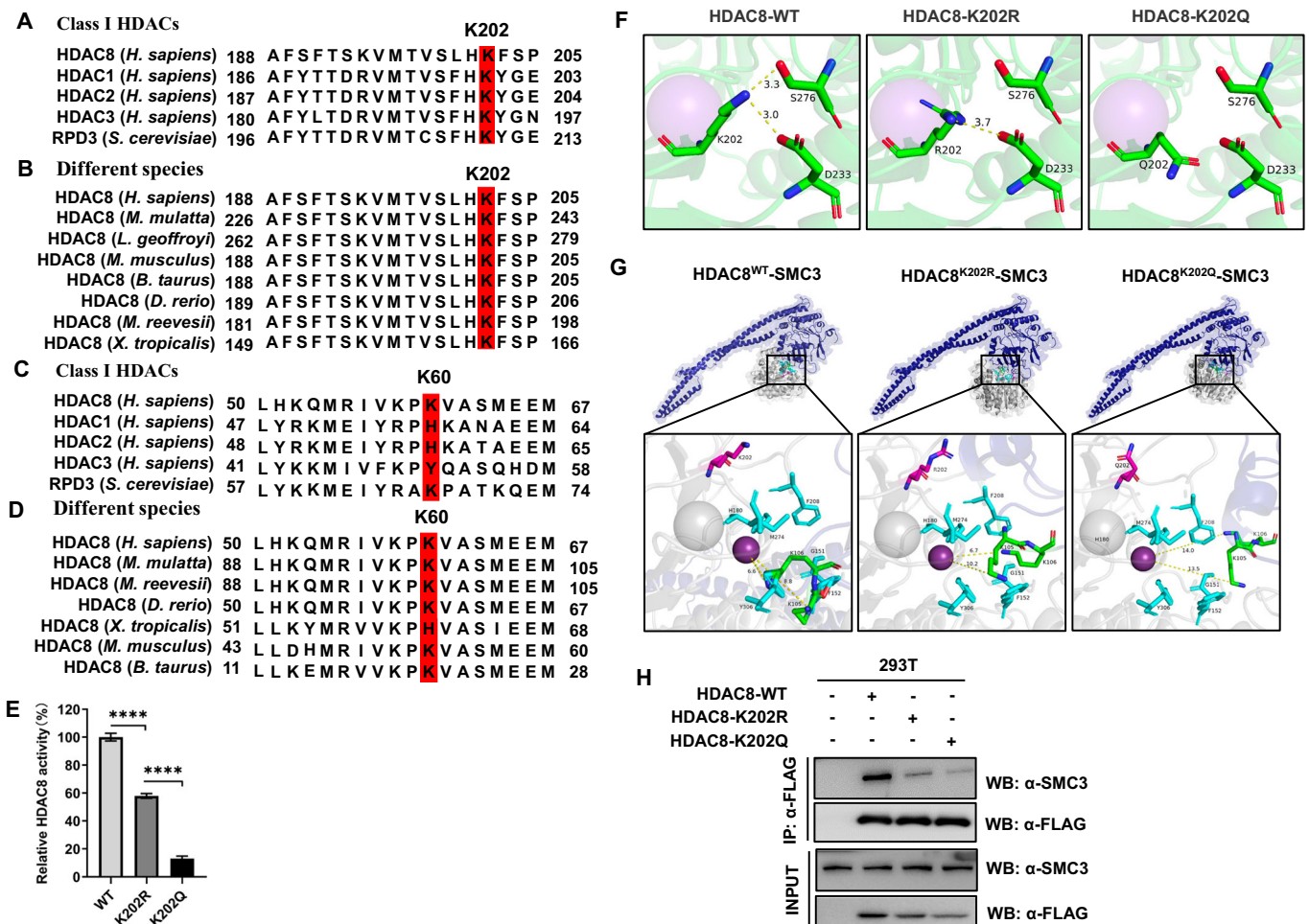

**Figure EV1. K202 acetylation disrupts HDAC8 structure and inhibits its activity.**

(A, B) K202 in HDAC8 is evolutionarily conserved. The sequences surrounding K202 in HDAC8 from different class I HDACs (A) and species (B) were aligned. (C, D) K60 in HDAC8 is not highly conserved. The sequences surrounding K60 in HDAC8 from different class I HDACs (C) and species (D) were aligned. (E) HDAC8-K202R and K202Q mutations lead to distinct reductions in deacetylase activity compared to WT protein. WT, K202R, and K202Q mutation of HDAC8 proteins were recombinant expressed E. coli, purified using nickel affinity chromatography, and then performed activity assay. Results were presented as the mean ± SD ($n = 3$, technical replicates). Differences between groups were compared using unpaired two-tailed Student's $t$ tests. ****$P < 0.001$. (F) HDAC8 K202Q mutation may disrupt the hydrogen bond network involving D233-K202-S276. The crystal structure of HDAC8-WT was obtained from previously published data (PDB accession code 1W22). The K202Q and K202R mutant structures of HDAC8 were predicted using the mutagenesis function of PyMOL based on the HDAC8-WT crystal structure. Dotted yellow lines indicated hydrogen bonds. In HDAC8-WT, robust hydrogen bond interactions were observed between K202-D233 (3.0 Å) and K202-S276 (3.3 Å). Conversely, in HDAC8-K202R, a hydrogen bond interaction was observed only between R202 and D233 (3.7 Å). In HDAC8 K202Q, Q202 failed to form hydrogen bonds with either D233 or S276. (G) Crystal structure of HDAC8 (WT, K202R, K202Q) -SMC3 complex predicted by ZDOCK 3.0.2 based on the crystal structure of HDAC8 (PDB accession code 1W22) and SMC3 (PDB accession code 7W1M). Compared to HDAC8-WT, the catalytic binding pockets of HDAC8-K202R and HDAC8 K202Q were increasingly distant from the acetylation sites K105 and K106 of SMC3, respectively. Overall structure of complex in PyMOL, with HDAC8 (WT, K202Q, K202R) and SMC3 shown as surfaces and cartoons. The key site (K202, Q202, R202), pocket walls (G151, F152, H180, F208, M274, Y306) of HDAC8 and the acetylation sites (K105, K106) of SMC3 in the focused panel were shown as sticks and colored in magenta, cyan and green respectively, Zinc ion was shown as a sphere colored in violet and the distance between Zinc ion and K105, K106 was shown. SMC3 was colored in density and HDAC8 (WT, K202Q, K202R) was colored in gray70. (H) Representative immunoblotting of 3 independent experiments shows that HDAC8 K202Q mutation causes diminished binding to the substrate SMC3. 293T cells were transfected with an empty vector, Flag-HDAC8-WT, Flag-HDAC8-K202R or Flag-HDAC8 K202Q. Whole-cell lysates were subjected to immunoprecipitation with anti-Flag beads, followed by immunoblot analysis to detect the precipitated proteins.

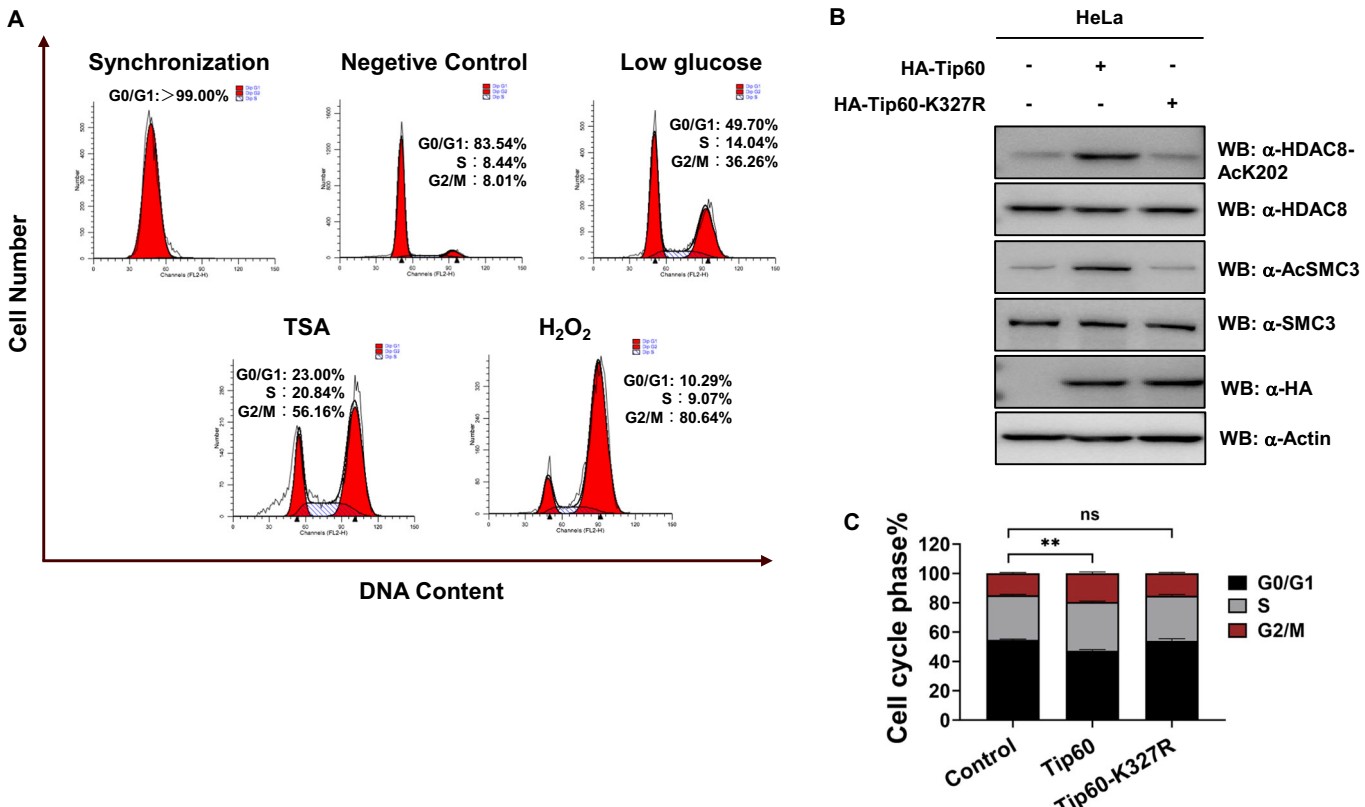

**Figure EV2. Overexpression of Tip60 causes elevated HDAC8 acetylation and cell cycle arrest.**

(**A**) Representative cell cycle distribution of HeLa cells synchronized in S-phase and treated with TSA, $H_2O_2$, glucose-free medium, or standard DMEM medium. Cells were initially synchronized in the early S-phase using a double-thymidine arrest and then exposed to TSA (500 nM), $H_2O_2$ (400 nM), glucose-free medium, or standard DMEM medium (as control) for 15 h. Flow cytometry analysis was performed with PI staining to detect cell cycle distribution. (**B, C**) Overexpression of Tip60 causes elevated acetylation levels of HDAC8 and SMC3 with concomitant G2/M phase arrest. HeLa cells were transfected with an empty vector, HA-Tip60 or HA-Tip60-K327R. Cells were harvested to prepare the whole-cell extracts for western blotting (**B**) or stain with PI for flow cytometry analysis (**C**). Results were presented as the mean ± SD ($n = 3$, biological replicates). Differences between groups were compared using unpaired two-tailed Student's $t$ tests. **$P < 0.01$; ns not significant.

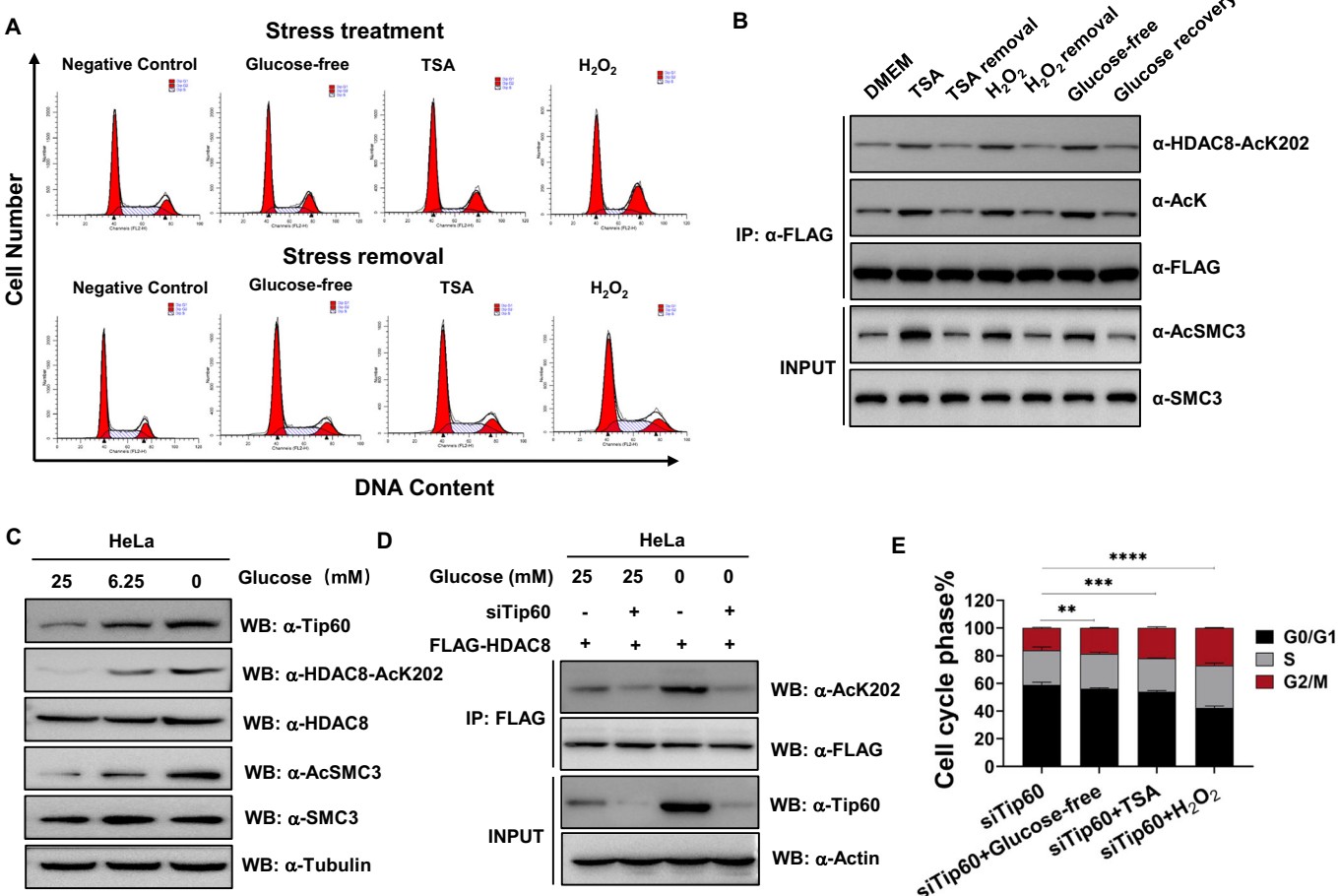

**Figure EV3.  K202 acetylation of HDAC8 is reversible.**

(A) Representative cell cycle distribution of HeLa cells after stress treatment and removal. HeLa cells were treated with TSA (500 nM), $H_2O_2$ (400 nM), glucose-free medium, or standard DMEM medium (as control) for 24 h, and then after removing these stresses, the cells were re-cultured in the standard DMEM medium for about 48 h. Cells were harvested to stain with PI for flow cytometry analysis after stress treatment and removal. (B) Representative immunoblotting of 3 independent experiments shows that stress-driven K202 acetylation of ectopically expressed HDAC8 is reversible in HeLa cells. Asynchronous HeLa cells overexpressing Flag-tagged HDAC8 were treated with TSA (500 nM), $H_2O_2$ (400 nM), glucose-free medium, or standard DMEM medium (as control) for 24 h and then after removing these stresses, the cells were re-cultured in the standard DMEM medium for about 48 h. Whole-cell lysates were subjected to immunoprecipitation with anti-Flag beads, followed by immunoblot analysis to detect the precipitated proteins. (C) Representative immunoblotting of 3 independent experiments shows that glucose starvation causes elevated expression of Tip60 and increased acetylation levels of HDAC8 and SMC3 in HeLa cells. Cells were treated with different concentrations of glucose, and then whole-cell extracts were prepared for immunoblot analysis. (D) Representative immunoblotting of 3 independent experiments shows that acetylation of HDAC8 is dependent on Tip60 with or without glucose-free stimulation. HeLa cells were transfected with Flag-HDAC8 and Tip60-siRNA, and then cultured with or without glucose condition. Whole-cell lysates were subjected to immunoprecipitation with anti-Flag beads, followed by immunoblot analysis to detect the precipitated proteins. (E) Simultaneous knockdown of Tip60 in stress-treated cells does not rescue cell cycle defects. HeLa cells transfected with Tip60-siRNA were treated with TSA (500 nM), $H_2O_2$ (400 nM), glucose-free medium, or standard DMEM medium (as control) for about 36 h. Cells were harvested for flow cytometry analysis. Results were presented as the mean ± SD ($n = 3$, biological replicates). Differences between groups were compared using unpaired two-tailed Student's $t$ tests. **$P < 0.01$; ***$P < 0.001$; ****$P < 0.0001$.

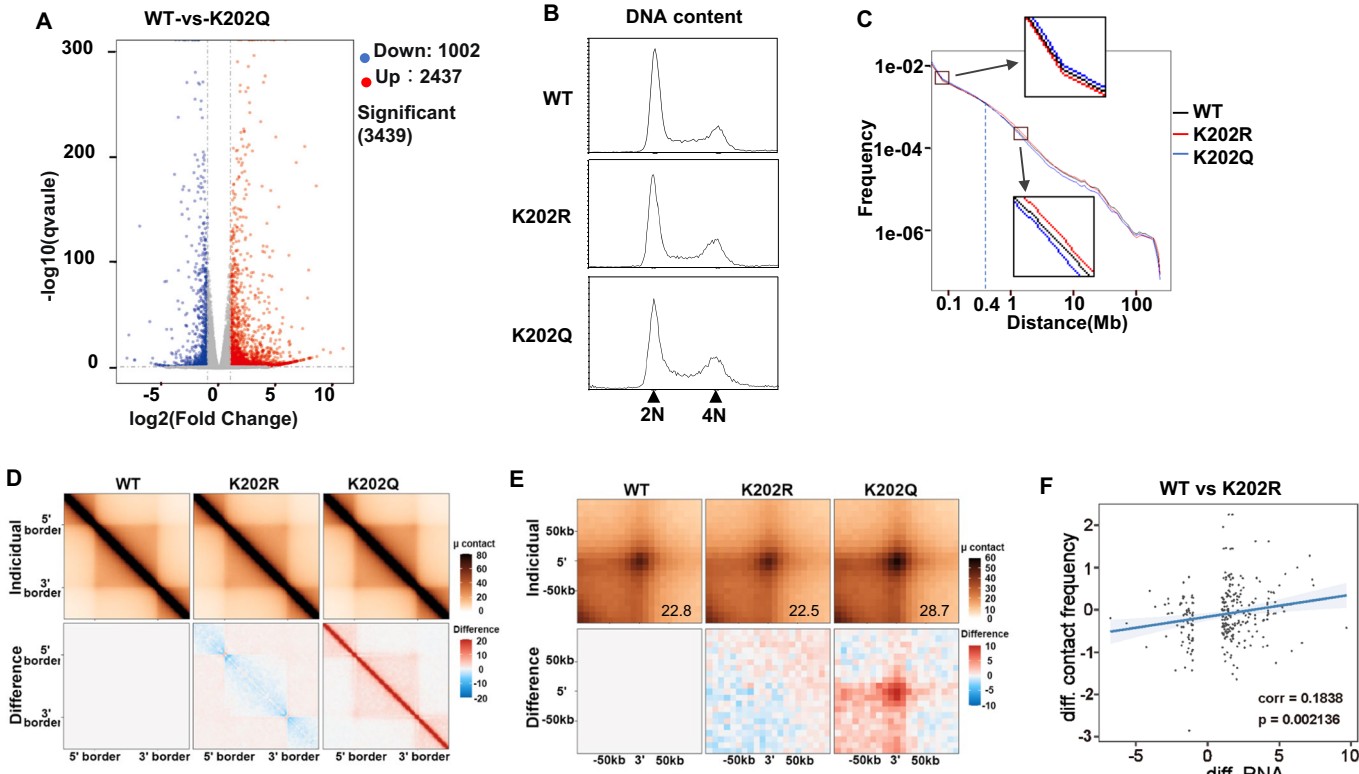

**Figure EV4. K2O2 acetylation promotes chromatin loop formation.**

(**A**) Volcano map of differentially expressed genes in the WT and K2O2Q cells. Significantly differentially expressed genes are shown as a red (up) or blue (down) dot. Non-significantly differentially expressed genes are shown as a gray dot. The horizontal and vertical coordinates represent the change of gene expression multiple and significance, respectively. (**B**) HeLa cells of WT, K2O2R, and K2O2Q mutation used for Hi-C experiments have similar cell cycle distributions as shown by flow cytometry analysis. (**C–E**) Another independent biological replicate of the Hi-C experiment for WT, K2O2R and K2O2Q HeLa cells, analyzed as in (**D–F**), respectively. Valid contacts were normalized to 56 million among the three samples. (**F**) Scatterplot for the difference of the number of identified significant chromatin interactions (*y* axis) and the difference in gene expression (*x* axis, merged by log2(FPKM + 1)) between WT and K2O2R cells. The blue solid line represents the fitted linear line, suggesting that the change of significant chromatin interactions positively correlates with the change in gene expression.

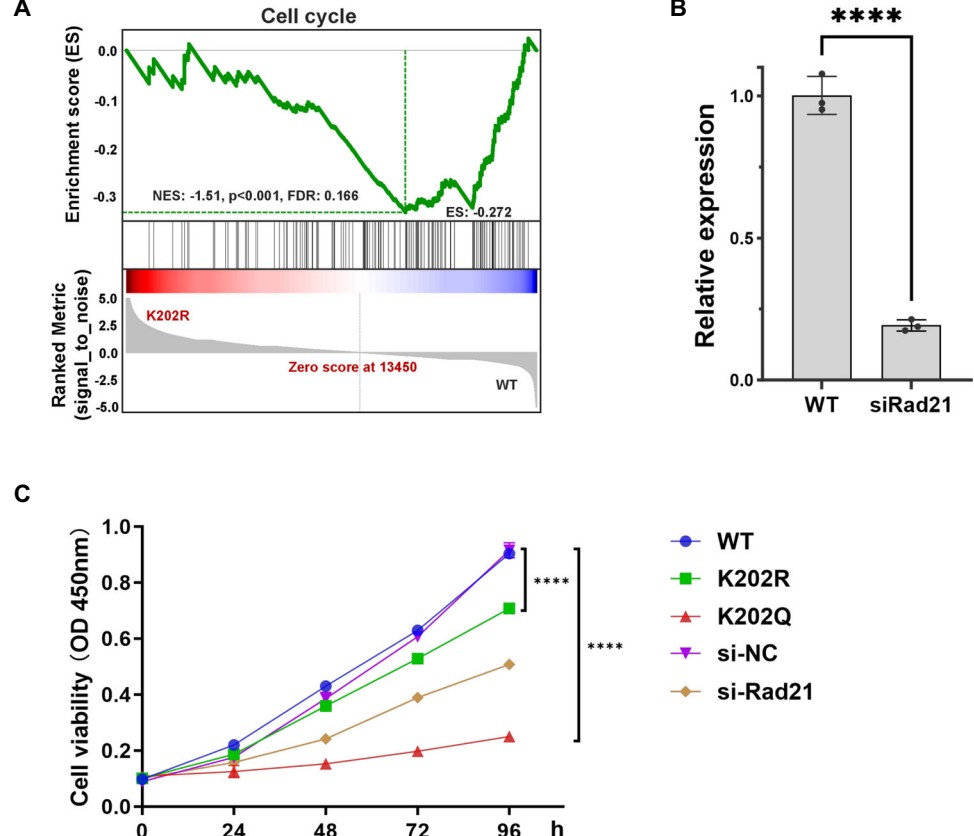

**Figure EV5.  K202 acetylation of HDAC8 inhibits cell proliferation.**

(**A**) Gene set enrichment analysis (GSEA) result of RNA-seq data in the HeLa cells of K202R mutation with WT. The cell cycle signaling pathway is enriched with an FDR of 0.166. (**B**) Knockdown efficiency in Rad21 expression by siRNA was determined by RT-qPCR. Rad21 relative expression levels were calculated using the ΔΔCt method and then normalized to the endogenous reference gene GAPDH. Results are presented as mean ± SD ($n = 3$, biological replicates). ****$P < 0.0001$. (**C**) HDAC8 K202Q mutation inhibits cell proliferation. Cell proliferation of WT, K202R, and K202Q cells was detected by CCK-8 assay, while HeLa cells transfected with siRNA for Rad21 knockdown were used as positive control. Results were presented as the mean ± SD ($n = 3$, technical replicates). Differences between groups were compared using unpaired two-tailed Student's $t$ tests. ****$P < 0.0001$.

