## [Peer Review File · EMBO Reports]

Reversible Acetylation of HDAC8 Regulates Cell Cycle

Chaowei Sang, Xuedong Li, Jingxuan Liu, Ziyin Chen, Minhui Xia, Miao Yu and Wei Yu

Corresponding Author(s): Wei Yu (yuw@fudan.edu.cn)

Review Timeline:

Submission Date:	14th Nov 23
Editorial Decision:	19th Dec 23
Revision Received:	21st Mar 24
Editorial Decision:	17th May 24
Revision Received:	28th May 24
Additional Correspondence:	25th Jun 24
Author's Response:	27th Jun 24
Accepted:	5th Jul 24

Editor: Deniz Senyilmaz Tiebe

Transaction Report:

Dear Prof. Yu,

Thank you for transferring your research manuscript to our journal, which was now seen by three referees, whose reports are copied below.

Referees express interest in the proposed role of HDAC8 acetylation in regulation of cell cycle. However, they also raise largely overlapping concerns that need to be addressed to consider publication here.

Should you be able to address the referee concerns satisfactorily, we would like to invite you to submit a revised manuscript. Please revise your manuscript with the understanding that the referee concerns (as in their reports) must be fully addressed and their suggestions taken on board. Please address all referee concerns in a complete point-by-point response. Acceptance of the manuscript will depend on a positive outcome of a second round of review. It is EMBO reports policy to allow a single round of major experimental revision only and acceptance or rejection of the manuscript will therefore depend on the completeness of your responses included in the next, final version of the manuscript.

We realize that it is difficult to revise to a specific deadline. In the interest of protecting the conceptual advance provided by the work, we recommend a revision within 3 months. Please discuss the revision progress ahead of this time with me if you require more time to complete the revisions, or if you have questions or comments regarding the revision (also by video chat).

1. A data availability section providing access to data deposited in public databases is missing (where applicable).
2. Your manuscript contains statistics and error bars based on $n=2$. Please use scatter plots in these cases.

You can submit the revision either as a Scientific Report or as a Research Article. For Scientific Reports, the revised manuscript can contain up to 5 main figures and 5 Expanded View figures, and it should not exceed 27000 characters. If the revision leads to a manuscript with more than 5 main figures it will be published as a Research Article. In this case the Results and Discussion section should be separate. If a Scientific Report is submitted, these sections have to be combined. This will help to shorten the manuscript text by eliminating some redundancy that is inevitable when discussing the same experiments twice. In either case, all materials and methods should be included in the main manuscript file.

4) a .docx formatted letter INCLUDING the reviewers' reports and your detailed point-by-point responses to their comments. As part of the EMBO publication's Transparent Editorial Process, EMBO reports publishes online a Review Process File (RPF) to accompany accepted manuscripts. This File will be published in conjunction with your paper and will include the referee reports, your point-by-point response and all pertinent correspondence relating to the manuscript.

<https://www.embopress.org/page/journal/14693178/authorguide#transparentprocess>

You are able to opt out of this by letting the editorial office know (emboreports@embo.org). If you do opt out, the Review Process File link will point to the following statement: "No Review Process File is available with this article, as the authors have

chosen not to make the review process public in this case."

5) a complete author checklist, which you can download from our author guidelines

<https://www.embopress.org/page/journal/14693178/authorguide>. Please insert information in the checklist that is also reflected in the manuscript. The completed author checklist will also be part of the RPF.

6) Please note that all corresponding authors are required to supply an ORCID ID for their name upon submission of a revised manuscript (<<https://orcid.org/>>). Please find instructions on how to link your ORCID ID to your account in our manuscript tracking system in our Author guidelines

<<https://www.embopress.org/page/journal/14693178/authorguide#authorshipguidelines>>

7) Before submitting your revision, primary datasets produced in this study need to be deposited in an appropriate public database (see <https://www.embopress.org/page/journal/14693178/authorguide#datadeposition>). Please remember to provide a reviewer password if the datasets are not yet public. The accession numbers and database should be listed in a formal "Data Availability" section placed after Materials & Method (see also

<https://www.embopress.org/page/journal/14693178/authorguide#datadeposition>). Please note that the Data Availability Section is restricted to new primary data that are part of this study. * Note - All links should resolve to a page where the data can be accessed. *

Additional information on source data and instruction on how to label the files are available:

<https://www.embopress.org/page/journal/14693178/authorguide#sourcedata>

9) Our journal encourages inclusion of *data citations in the reference list* to directly cite datasets that were re-used and obtained from public databases. Data citations in the article text are distinct from normal bibliographical citations and should directly link to the database records from which the data can be accessed. In the main text, data citations are formatted as follows: "Data ref: Smith et al, 2001" or "Data ref: NCBI Sequence Read Archive PRJNA342805, 2017". In the Reference list, data citations must be labeled with "[DATASET]". A data reference must provide the database name, accession number/identifiers and a resolvable link to the landing page from which the data can be accessed at the end of the reference. Further instructions are available at <http://www.embopress.org/page/journal/14693178/authorguide#referencesformat>

10) Regarding data quantification (see Figure Legends:

<https://www.embopress.org/page/journal/14693178/authorguide#figureformat>)

12) Please also note our reference format:

I look forward to seeing a revised version of your manuscript when it is ready. Please let me know if you have questions or comments regarding the revision.

Kind regards,

Deniz Senyilmaz Tiebe

Deniz Senyilmaz Tiebe, PhD
Scientific Editor
EMBO Reports

Referee #1:

This is a paper that attempts to find the acetylation of HDAC8 and investigate its physiological function. Hdac8 is a cohesin deacetylase that is important for regulation of cohesin dynamics, and thus regulates cohesin function in cell cycle and transcription. I think this paper has an impact, but I am concerned about many points as follows. For example, in Fig 1f, they used a system in which HDAC8 expressed in *E. coli* is acetylated in *E. coli*, and it shows that the activity is reduced by K202ac. It certainly shows acetylation in K202, but acetylation of other residues must also be occurring. To prove that acetylation of K202ac reduces the activity of HDAC8, it is necessary to perform the same experiment with K202R. Fig4: They have data showing that the interaction in the TAD is stronger in cells heterozygous for K202Q (acetyl mimic) and that the loop gets also stronger. This mutation reduces the activity of HDAC8 (Fig1c). This means that the cohesin pool for loop extrusion is reduced. As far as I understand from other papers, this is strange. To resolve this discrepancy, I would like authors to do a basic check of the cells in which K202Q was introduced into the genome: what is the abundance of cohesin, what is the abundance in the chromatin fraction, what is the abundance of Smc3ac, etc. Fig5, d&e; the K202Q mutant cell line has a cohesion defect of over 70%. I wonder if such a high level of cohesion defects can cause death. I think that authors should perform a cohesion assay with a control such as siRad21 in parallel to see if there are any qualitative or quantitative differences. I raised here the most important points I feel that should be addressed at least. Overall the data contains lots of roughness. Therefore I could not recommend this paper to be published in any journal.

Referee #2:

Cohesin is a complex with a ring-shaped structure ensuring sister chromatid cohesion as well as DNA looping. Cohesin association with DNA is regulated by different co-factors. Loading relies Scc2 while unloading depends on a protein called Wpl1. Unloading activity unloads cohesin from DNA by opening a gate between Scc1 and Smc3. To stabilize sister chromatid from S to M phase, the unloading activity is repressed during S phase by Eco1-mediated acetylation of a pair of conserved lysine residues (K112/113 in yeast and K105/106 in human) within the Smc3 ATPase head. Smc3 acetylation is maintained throughout the G2 and M phases and only removed at anaphase onset by a deacetylase called Hos1/HDAC8. It is thought that Hos1/HDAC8 mediated Smc3-deacetylation is important to recycle Smc3. Recent study in human cells also suggested that HDAC8 mediated Smc3 deacetylation regulates expansion of DNA loop during interphase (van Ruiten et al. 2023). Despite these interesting findings little is known about mechanisms regulated HDAC8 activity. In the present study the authors claim that acetylation of K202 in HDAC8 as a critical event that exerts a negative modulatory influence on its enzymatic activity. Overall, the manuscript is clearly written, of good technical quality and shows important results. However, I would recommend accepting it only after addressing some of the following points:

Major comments

There are some key references that should be mentioned in the introduction, in particular the pioneered works that described the Smc3 acetylation cycle (Unal et al 2008, Rolef et al 2008 and Zhang et al; 2008). It is worth mentioning the studies that demonstrated for the first time that Hos1/HDAC8 is the enzyme that deacetylates Smc3 during anaphase (Borges et al 2010, Beckouet et al 2010 and Xiong et al. 2010). The work from van Ruiten et al. 2023 should be also mentioned in the introduction.

Figure 1. The authors provide a set of in vivo and invitro data demonstrating that HDAC8 is acetylated at lysine K202. Based on these they concluded that acetylation of residue K202 has the effect of inhibiting deacetylation of Smc3 by HDAC8. However,

they did not determine clearly the impact of the K202 mutation on Smc3 acetylation in human cells. The authors performed CRISPR mutations in the HDAC8 gene but surprisingly did not study the impact of these mutations on Smc3 acetylation. To make sure that K202 acetylation inhibits HDAC8 mediated Smc3 de-acetylation in living cells it is necessary to analyze the impact of these mutations on Smc3 acetylation in a revised version of the article.

Figure 2

Figure 2C. Based on the fact that Tip60 overexpression induces an increase in Smc3 and HDAC8 acetylation, the authors claim that Tip60 is the acetylase that induces HDAC8 acetylation. However, it is conceivable that Tip60 overexpression causes cell arrest in G2/M, which in turn induces increased acetylation of Smc3 and HDAC8. It is therefore important to monitor the effect of Tip60 overexpression on the cell cycle. The authors should also monitor the impact of Tip60 overexpression on the cellular amount of HDAC8 in their western blot.

Figure 3D. The authors should repeat experiment Figure 3D in a context where cells express K202Q and K202R mutations.

Figure 4. It would have been interesting to compare the effect of the K202 mutations with that induced by inactivation of the HDAC8 gene. Rowland's Benjamin laboratory recently studied the effect of the absence of HDAC8 on genome organization in HAP1 cells. Would it be possible for the authors to use this same type of cell to study the effect of their mutations on genome organisation? This would also make it possible to compare the effects of HDAC8 point mutations with those induced by the absence of HDAC8.

Minor comment:

The Hi-C maps should be square and not rectangular. The color code must be added next to the Hi-C maps.

Referee #3:

HDAC8 is a class I deacetylase that deacetylates the cohesin complex subunit SMC3. How HDAC8's deacetylase activity is controlled is poorly understood. Sang et al. find that HDAC8 is regulated by TIP60-mediated acetylation of K202 on HDAC8. K202 is positioned close to the active site and they find that acetylation of this lysine inhibits HDAC8's deacetylase activity. They show that K202 acetylation is dynamic during the cell cycle. Using mutants that either mimic the acetylated lysine, or the non-acetylated lysine, they find that acetylation of this site alters gene expression and regulates the 3D genome. The findings presented in this manuscript are clearly interesting, but further support is required for the model put forward by the authors to be suitable for publication in EMBO Reports.

Comments Sang et al.

- The authors generated cell lines harbouring acetylation mimick and non-acetylatable mutants for K202, to assess the effect on gene expression and the 3D genome. When preparing their samples for Hi-C analyses, they comment that these cell lines have a similar cell cycle stage and can therefore be compared. However, in the next figure they show that mutation of K202 leads to an arrest in G2/M-phase and changes in gene expression. It would therefore be important to perform Hi-C analyses in synchronized cells, preferably G1-phase to prevent any potential influence by changes in cohesin complexes holding the sister chromatids together. It would also be important to check the levels of HDAC8 in these cells and assess whether the acetylation of SMC3 is affected.
- The authors should include replicates for the Hi-C analyses, preferably in independent clones. They should also provide information on the amount of reads in the methods and whether these are similar between genotypes. The differences observed in for example figure 4e could (at least partially) be explained by a difference in sequencing depth.
- It appears to me that the changes depicted in figure 4g and 4h are rather small. Is this considered a strong correlation? The main text describing these panels is rather limited and could use further explanation.
- The finding that K202Q mutant cells display a strong cohesion defect is interesting. To draw meaningful conclusions, it would be important to score the severity of the cohesion defect and provide example pictures of the different categories.
- The authors show in figure 2e and 3b that several stressors lead to a G2/M arrest. However, the extent to which this arrest occurs is very different between these two panels. Could the authors explain these differences?
- In figure 3c the authors show that cells can recover from these stressors and that both the acetylation of HDAC8 and SMC3 is rescued to normal wild type unperturbed levels. However, it is difficult to compare these settings, as the western blot samples are harvested from cells that are from different cell cycle phases, and SMC3 acetylation is known to change throughout the cell cycle. The increase in acetylated SMC3 observed in stressed cells could therefore be at least partially explained by the enrichment of cells in G2 phase. It would be important to compare the acetylation between e.g. only G1 cells or G2 cells. In

addition the authors could test whether the increase in acetylated SMC3 in stressed cells is dependent on HDAC8 acetylation by simultaneously depleting Tip60, and whether this rescues the cell cycle defect.

- The finding that Tip60 can interact with HDAC8 and acetylate it, is clear from figure 2. Is this acetyl transferase only active when stressors are present, and/or is Tip60 also cell cycle regulated? It would be useful for the reader if Tip60 is better introduced in the text.
- The finding that K202R also shows a decrease in its deacetylation is surprising. The authors hypothesise that this K202 might be important for the catalytic activity. Could they further elaborate on how K202 might do so?
- Could the authors elaborate on how the acetylation of K202 on HDAC8 reduces the deacetylase activity? Does acetylation of K202 e.g. change the folding of this part of HDAC8? Does it prevent entry of acetylated lysines into the deacetylation pocket?
- It is unclear to me how the authors assessed the total amount of acetylation on HDAC8 in figure 2d. Did the authors first enrich for HDAC8 with a pulldown? Or does that this antibody recognize multiple acetylated residues on HDAC8?
- It would be informative if the authors include Hos1, the *S. cerevisiae* cohesin deacetylase, in their evolutionary conservation alignments.
- The authors should include loading controls for all their western blot analyses.
- The authors should always show both the input and the pulldown for the IP experiments they perform. E.g. in figure 1g they only show the pulldown, but not the levels of AcK202, AcK, and FLAG in the input. In figure 2a and 2b, please show both the signal for FLAG and HA in the input.

Dear Editor and reviewers,

Thank you for your message, and we extend our gratitude to all reviewers for their diligent efforts in reviewing our manuscript. We highly value the positive comments from you and the reviewers, and we are pleased to report that we have completed the three months of experiments. We are confident that our revisions will substantially enhance the quality of this work in response to the reviewers' concerns. With this, we are submitting the revised manuscript, which meticulously addresses each of the reviewers' points.

Referee #1: Cohesin is a complex with a ring-shaped structure ensuring sister chromatid cohesion as well as DNA looping. Cohesin association with DNA is regulated by different co-factors. Loading relies Scc2 while unloading depends on a protein called Wpl1. Unloading activity unloads cohesin from DNA by opening a gate between Scc1 and Smc3. To stabilize sister chromatid from S to M phase, the unloading activity is repressed during S phase by Eco1-mediated acetylation of a pair of conserved lysine residues (K112/113 in yeast and K105/106 in human) within the Smc3 ATPase head. Smc3 acetylation is maintained throughout the G2 and M phases and only removed at anaphase onset by a deacetylase called Hos1/HDAC8. It is thought that Hos1/HDAC8 mediated Smc3-deacetylation is important to recycle Smc3. Recent study in human cells also suggested that HDAC8 mediated Smc3 deacetylation regulates expansion of DNA loop during interphase (van Ruiten et al. 2023). Despite these interesting findings little is known about mechanisms regulated HDAC8 activity. In the present study the authors claim that acetylation of K202 in HDAC8 as a critical event that exerts a negative modulatory influence on its enzymatic activity. Overall, the manuscript is clearly written, of good technical quality and shows important results. However, I would recommend accepting it only after addressing some of the following points:

We thank the reviewer for the positive comments and insightful suggestions. Please find our point-to-point response below.

Major comments

There are some key references that should be mentioned in the introduction, in particular the pioneered works that described the Smc3 acetylation cycle (Unal et al 2008, Rolef et al 2008 and Zhang et al; 2008). It is worth mentioning the studies that demonstrated for the first time that Hos1/HDAC8 is the enzyme that deacetylates Smc3 during anaphase (Borges et al 2010, Beckouet et al 2010 and Xiong et al. 2010). The work from van Ruiten et al. 2023 should be also mentioned in the introduction.

Response: We appreciate the reviewer's suggestion and have added the relevant references mentioned above to the introduction in the revised version.

Figure 1. The authors provide a set of in vivo and invitro data demonstrating that HDAC8 is acetylated at lysine K202. Based on these they concluded that acetylation of residue K202 has the effect of inhibiting deacetylation of Smc3 by HDAC8. However, they did not determine clearly the impact of the K202 mutation on Smc3 acetylation in human cells. The authors performed CRISPR mutations in the HDAC8 gene but surprisingly did not study the impact of these mutations on Smc3 acetylation. To make sure that K202 acetylation inhibits HDAC8 mediated Smc3 de-acetylation in living cells it is necessary to analyze the impact of these mutations on Smc3 acetylation in a revised version of the article.

Response: We appreciate the valuable suggestion from the reviewer. Following your guidance, we conducted Western blot analysis on Knock-in HDAC8 K202R and K202Q cells (New Figure 4b). Our findings reveal a noteworthy increase in the acetylation level of SMC3 in HeLa cells subsequent to the introduction of HDAC8 K202R and K202Q mutations.

New Figure 4b

Figure for referee with unpublished data has been removed upon request by the authors.

Figure 2C. Based on the fact that Tip60 overexpression induces an increase in Smc3 and HDAC8 acetylation, the authors claim that Tip60 is the acetylase that induces HADC8 acetylation. However, it is conceivable that Tip60 overexpression causes cell arrest in G2/M, which in turn induces increased acetylation of Smc3 and HADC8. It is therefore important to monitor the effect of Tip60 overexpression on the cell cycle. The authors should also monitor the impact of Tip60 overexpression on the cellular amount of HDAC8 in their western blot.

Response: We thank the reviewer for this suggestion. Following your suggestions, we ectopically overexpressed Tip60 in HeLa cells with approximately 30-40% confluence for 24h, and then analyzed cell cycle stages by FACS as well as monitored intracellular HDAC8 amount by immunoblotting. In the figure below, the impact of Tip60 overexpression on the cell cycle was obvious (Supplementary Figure 2b), and the intracellular HDAC8 amount remained generally stable while the acetylation levels of HDAC8 and SMC3 were significantly increased (Supplementary Figure 2c). These findings suggested that Tip60 could affect the cell cycle through the acetylation of HDAC8.

However, several of our additional experiments have provided evidence that Tip60 is capable of acetylating HDAC8. As depicted in Figure 2a, b, it clearly demonstrated the interaction between Tip60 and HDAC8. Through knockdown Tip60 by using siRNA, we established Tip60's ability to deacetylate HDAC8 (Figure 2d). Further experiments

indicated that acetylation of HDAC8-K202 are dependent on Tip60, regardless of the presence or absence of stress stimulation (Supplementary Figure 3d).

New Supplementary Figure 2b-2c

Figure for referee with unpublished data has been removed upon request by the authors.

Figure 2a-b

Figure for referee with unpublished data has been removed upon request by the authors.

New Figure 2d

New Supplementary Figure 3d

Figure for referee with unpublished data has been removed upon request by the authors.

Figure 3D. The auteurs should repeat experiment Figure 3D in a context where cells express K202Q and K202R mutations.

Response: Thank you for the question. In original Figure 3d, we executed cell synchronization and subsequently examined the endogenous acetylation changes at

K202 of HDAC8 throughout the cell cycle progression. Therefore, we were unable to detect the K202Q/R mutants within the endogenous cell population.

Original Figure 3d

Figure for referee with unpublished data has been removed upon request by the authors.

Figure 4. It would have been interesting to compare the effect of the K202 mutations with that induced by inactivation of the HDAC8 gene. Rowland's Benjamin laboratory recently studied the effect of the absence of HDAC8 on genome organization in HAP1 cells. Would it be possible for the authors to use this same type of cell to study the effect of their mutations on genome organisation? This would also make it possible to compare the effects of HDAC8 point mutations with those induced by the absence of HDAC8.

Response: We appreciate the valuable suggestion from the reviewer. It is worth noting that HAP1 cells are haploid, whereas HeLa cells are hyper-triploid, posing challenges in gene editing for HeLa cells. Despite this, we chose HeLa cells for our research due to their suitability for studying the cell cycle dynamics. The work by Rowland et al. implies a positive regulation of loop formation by HDAC8, where HDAC8 knockdown enhances PDS5A binding to acetylated SMC3, resulting in looping defects and reduced chromatid loop formation (the figure in van Ruiten et al. 2023). In contrast, our study indicates a negative regulation of loop formation by HDAC8. Our findings show that HDAC8 point mutations significantly enhance chromatid loop formation (Figure 4f in our study). It is noteworthy that another recently published paper by Kota Nagasaka et al. (2023, Molecular Cell, PMID: 37591243) found that treating HeLa cells with an HDAC8 inhibitor (PCI) also promotes loop formation (the figure in Kota Nagasaka et al. 2023, PMID: 37591243). Interestingly, this paper provides additional support to our findings. We hypothesize that differences in cell types or experimental techniques may underlie the varying regulatory effects of HDAC8 on loop formation observed across these studies.

van Ruiten et al. 2023

Our study (Figure 4f)

Figures for referee with unpublished data have been removed upon request by the authors.

Kota Nagasaka et al. 2023

Figures for referee with unpublished data have been removed upon request by the authors.

Minor comment: The Hi-C maps should be square and not rectangular. The color code must be added next to the Hi-C maps.

Response: We appreciate the reviewer's suggestion and have corrected these in the revised version.

New Figure 4i

Figure for referee with unpublished data has been removed upon request by the authors.

Referee #2: HDAC8 is a class I deacetylase that deacetylates the cohesin complex subunit SMC3. How HDAC8's deacetylase activity is controlled is poorly understood. Sang et al. find that HDAC8 is regulated by TIP60-mediated acetylation of K202 on HDAC8. K202 is positioned close to the active site and they find that acetylation of this lysine inhibits HDAC8's deacetylase activity. They show that K202 acetylation is dynamic during the cell cycle. Using mutants that either mimic the acetylated lysine, or the non-acetylated lysine, they find that acetylation of this site alters gene expression and regulates the 3D genome. The findings presented in this manuscript are clearly interesting, but further support is required for the model put forward by the authors to be suitable for publication in EMBO Reports.

We thank the reviewer for the positive comments and insightful suggestions. Please find our point-to-point response below.

Comments Sang et al.

-The authors generated cell lines harbouring acetylation mimick and non-acetylatable mutants for K202, to assess the effect on gene expression and the 3D genome. When preparing their samples for Hi-C analyses, they comment that these cell lines have a similar cell cycle stage and can therefore be compared. However, in the next figure they show that mutation of K202 leads to an arrest in G2/M-phase and changes in gene expression. It would therefore be important to perform Hi-C analyses in synchronized cells, preferably G1-phase to prevent any potential influence by changes in cohesin complexes holding the sister chromatids together. It would also be important to check the levels of HDAC8 in these cells and assess whether the acetylation of SMC3 is affected.

Response: We appreciate the reviewer's valuable suggestion. The analysis of unsynchronized cells with the K202Q mutation for Hi-C experiments, as illustrated in the supplementary Figure 4b, revealed a slightly increased proportion of cells in the G2/M phase in K202Q cells, consistent with the statistical findings presented in Figure 5f. However, we considered this difference acceptable and emphasized that the cell samples had comparable cell cycle stages. While using synchronized G1 phase cells for Hi-C experiments would be an ideal choice, our attempts to synchronize cells to the G1 phase using thymidine arrest, especially in the K202Q mutant cells (Results of attempted synchronization to G1 phase), proved challenging due to varied proliferation statuses among mutant cells (Supplementary figure 5c). Achieving strictly synchronized G1-phase cells may be impractical. Therefore, we chose to utilize unsynchronized cells, which offered the advantage of reduced variability in cell cycle stage. It's worth noting that recent single-cell Hi-C analyses have indicated that loops, as more conserved structures, remain largely

unchanged throughout the cell cycle (Nagano et al. 2017 Nature, PMID: 28682332). Additionally, findings from other studies, such as the observed promotion of loop formation in unsynchronized HeLa cells treated with an HDAC8 inhibitor (Kota Nagasaka et al. 2023 Molecular Cell, PMID: 37591243), align with our results. Hence, slight variations in G2/M phase distribution among Hi-C samples are unlikely to significantly impact our conclusions. Moreover, the integrated analysis of RNA-seq and Hi-C results on unsynchronized cells allows for linking chromatin loops to gene expression changes, providing comprehensive insights into regulatory mechanisms. Additionally, in response to the reviewer's suggestion, we conducted a basic immunoblotting examination of the knock-in cells (Figure 4b). As expected, the SMC3 acetylation level was significantly elevated in the K202Q cells, while the HDAC8 expression level remained essentially unchanged compared to WT cells. This further supports our findings and strengthens the correlation between HDAC8 activity and chromatin loop dynamics.

New Supplementary Figure 4b Results of attempted synchronization to G1 phase

Figure for referee with unpublished data has been removed upon request by the authors.

New Supplementary Figure 5c

New Figure 4b

Figures for referee with unpublished data have been removed upon request by the authors.

-The authors should include replicates for the Hi-C analyses, preferably in independent clones. They should also provide information on the amount of reads in the methods and whether these are similar between genotypes. The differences observed in for example figure 4e could (at least partially) be explained by a difference in sequencing depth.

Response: We appreciate the reviewer's constructive suggestion. In response to your guidance, we conducted a replicative Hi-C experiment using an additional site mutant clone. To mitigate potential biases arising from sequencing depth, we normalized valid contacts to ensure consistency among the three samples. The subsequent analysis produced the results presented in the below figure. Additionally, to enhance transparency and facilitate

further scrutiny, we have deposited the high-throughput sequencing data from both independent Hi-C experiments into the GEO database. This step aims to provide open access to the raw data, promoting reproducibility and allowing other researchers to explore and validate our findings.

New Figure 4d-f: Replicate 1 (normalized to **63 million** valid contacts per sample):

Figure for referee with unpublished data has been removed upon request by the authors.

New Supplementary Figure 4c-e: Replicate 2 (normalized to **56 million** valid contacts per sample):

Figure for referee with unpublished data has been removed upon request by the authors.

-It appears to me that the changes depicted in figure 4g and 4h are rather small. Is this considered a strong correlation? The main text describing these panels is rather limited and could use further explanation.

Response: We appreciate the reviewer's comments. Gene expression is a highly intricate process, and the spatiotemporal aspects of genomic structure are increasingly acknowledged as crucial for understanding eukaryotic gene expression. Nevertheless, the mechanistic underpinnings and causal links between structure and gene expression remain poorly understood. The development and application of Hi-C technology have provided valuable insights into such studies. In our study, we employed multi-omics analysis to investigate the relationship between chromatin interactions and gene expression. Initially, we selected differentially expressed genes (DEGs) with more than 2-fold changes between K202Q/K202R with WT cells from the RNA-seq data. Meanwhile, we obtained normalized contact frequencies of chromatin loops from Hi-C data. By defining the ± 1 kb regions around the transcription start site (TSS) of genes as the approximate promoter regions, we overlapped the promoters of DEGs with chromatin loops to identify loops associated with these genes (generated .csv file had been submitted as source data with the title as shown below). Subsequently, we calculated the Pearson correlation

coefficient (PCC) and the corresponding p-value between the changes in gene expression and the changes in normalized contact frequency in chromatin loops (Figure 4g, h). Through the overall analysis, we discovered that the normalized contact frequency of interactions exhibited a significant correlation with gene expression levels at the distal elements of promoters. Our findings underscore the pivotal role of HDAC8-mediated chromatin interactions in regulating gene expression. Given the intricate nature of gene expression regulation, the significant correlation shown in our overall analysis provides robust support for our conclusions. Also, following the comments, we have corrected the “strong correlation” to “significant correlation” and added further explanation in the revised manuscript.

Figure 4g-4h

Figure for referee with unpublished data has been removed upon request by the authors.

-The finding that K202Q mutant cells display a strong cohesion defect is interesting. To draw meaningful conclusions, it would be important to score the severity of the cohesion defect and provide example pictures of the different categories.

Response: We appreciate the reviewer's suggestion, and in response, we have implemented additional improvements to our study. Specifically, we further classified the

observed cohesion phenotypes into four categories: normal, mild, moderate, and severe, as illustrated in Figure 5e. Additionally, we conducted a chromosome spread assay with siRad21 as a positive control, clearly demonstrating the knockdown effect of Rad21, as depicted in Supplementary Figure 5b. Results were shown in the Figure 5d. These refinements aim to provide a more detailed and comprehensive analysis of cohesion phenotypes and strengthen the overall robustness of our experimental results.

New Figure 5e

New Supplementary Figure 5b

Figure for referee with unpublished data has been removed upon request by the authors.

New Figure 5d

Figure for referee with unpublished data has been removed upon request by the authors.

-The authors show in figure 2e and 3b that several stressors lead to a G2/M arrest. However, the extent to which this arrest occurs is very different between these two panels. Could the authors explain these differences?

Response: Thank you for providing clarification regarding the differences in experimental conditions between Figure 2e and Figure 3b. The distinction in the synchronization status of HeLa cells, with Figure 2e involving cells synchronized to the early S phase and Figure 3b utilizing unsynchronized cells, explains the observed variations between these two panels. We have added this information into figure legends to help the interpretation of the results presented in the figures.

-In figure 3c the authors show that cells can recover from these stressors and that both the acetylation of HDAC8 and SMC3 is rescued to normal wild type unperturbed levels. However, it is difficult to compare these settings, as the western blot samples are harvested from cells that are from different cell cycle phases, and SMC3 acetylation is known to change throughout the cell cycle. The increase in acetylated SMC3 observed in stressed cells could therefore be at least partially explained by the enrichment of cells in G2 phase. It would be important to compare the acetylation between e.g. only G1 cells or G2 cells. In addition, the authors could test whether the increase in acetylated SMC3 in stressed cells is dependent on HDAC8 acetylation by simultaneously depleting Tip60, and whether this rescues the cell cycle defect.

Response: We thank the reviewer's insightful suggestion. As shown in Figure 3d, we conducted overexpression of Flag-HDAC8 in HeLa cells followed by stress treatment. Subsequently, we enriched the HDAC8 protein using Flag beads and measured the enzyme activity in vitro, confirming a significant reduction in HDAC8 enzyme activity induced by stress. Extensive research has established SMC3 as an unequivocal substrate of HDAC8, and inhibition of HDAC8 has been linked to elevated intracellular SMC3 acetylation levels. Furthermore, in our experiments, cells treated with stress were not restricted to G1/G2 phases post-cell cycle synchronization. Although stress-induced G2/M phase arrest was statistically significant, the overall alteration was relatively minor (Figure 3b). Moreover, the duration of TSA treatment and glucose starvation (24-36 hours) exceeded the typical cell cycle duration, rendering it challenging to obtain G1/G2 cells post-cell cycle synchronization following stress. In summary, we posit that the observed increase in SMC3 acetylation levels following stress treatment primarily arises from the reduction in HDAC8 enzyme activity.

New Figure 3d

Figure 3b

Figures for referee with unpublished data have been removed upon request by the authors.

Following your recommendations, we proceeded with Tip60 knockdown in stress-stimulated cells and subsequently evaluated the cell cycle stage using FACS analysis. However, despite the Tip60 knockdown, we observed no rescue of the cell cycle defect (Supplementary Figure 3e). Additionally, we examined acetylated K202-HDAC8 levels in glucose-starved cells with Tip60 knockdown (Supplementary Figure 3d). Notably, while Tip60 protein levels increased under glucose-free stimulation (line 3 compared to line 1), the acetylated K202-HDAC8 returned to normal levels under Tip60 knockdown conditions despite glucose starvation (line 4 compared to line 2). These findings suggest that the stimulations triggering increased acetylation of K202-HDAC8 are dependent on Tip60.

New Supplementary Figure 3e New Supplementary Figure 3d

Figures for referee with unpublished data have been removed upon request by the authors.

- The finding that Tip60 can interact with HDAC8 and acetylate it, is clear from figure 2. Is this acetyl transferase only active when stressors are present, and/or is Tip60 also cell cycle regulated? It would be useful for the reader if Tip60 is better introduced in the text.

Response: We appreciate the valuable suggestion from the reviewer. In line with your recommendation, we investigated the expression of Tip60 throughout the cell cycle, revealing that the expression level of Tip60 remained unchanged in synchronized cells (Figure 3e). This finding suggests that other regulatory mechanisms, such as posttranslational modifications, may govern Tip60 activity during the cell cycle.

New Figure 3e

Figure for referee with unpublished data has been removed upon request by the authors.

Furthermore, we investigated the alterations in Tip60 expression under various glucose concentration treatments (Supplementary Figure 3c). Our results demonstrated an increase in Tip60 expression under conditions of decreased glucose concentration, consistent with previous findings (2019 Cell Rep, PMID: 30699357). This observation suggests that glucose starvation may upregulate Tip60 expression, thereby influencing HDAC8 acetylation.

New Supplementary Figure 3c

Figure for referee with unpublished data has been removed upon request by the authors.

-The finding that K202R also shows a decrease in its deacetylation is surprising. The authors hypothesise that this K202 might be important for the catalytic activity. Could they further elaborate on how K202 might do so?

Response: We have consolidated our responses to this and the subsequent comment. Please refer to the response provided for the next question.

- Could the authors elaborate on how the acetylation of K202 on HDAC8 reduces the deacetylase activity? Does acetylation of K202 e.g. change the folding of this part of HDAC8? Does it prevent entry of acetylated lysines into the deacetylation pocket?

Response: According to Decroos et al. (2015, Biochemistry, PMID: 26463496), K202 plays a crucial role in establishing a hydrogen bond network (D233-K202-S276) essential for HDAC8 activity. Specifically, in wild-type (WT) HDAC8, the side chain of K202 forms hydrogen bonds concurrently with S276 and D233 (the left figure in Decroos et al. 2015). Mutation of D233 to G233 results in a significant decrease in protein thermostability ($\Delta T_m = -6.8^\circ\text{C}$) and only 49% residual activity. Crystal structure analysis reveals that the G233 mutation induces minor structural changes but disrupts the hydrogen bond with K202 and weakens the remaining hydrogen bond between S276 and K202 (the right figure in Decroos et al. 2015). Further molecular dynamics (MD) simulations demonstrate increased root mean square (rms) fluctuations of K202 and adjacent residues upon the G233 mutation. Given that K202 resides at the end of the β -strand, S276 is positioned in the L7 loop, and D233 is located in the L6 loop, the hydrogen bond network involving D233-K202-S276 is deemed critical for stabilizing the HDAC8 structure. In our study, we observed that the K202R mutation disrupted the hydrogen bond with S276 and weakened the bond with D233, while the K202Q mutation led to the simultaneous loss of hydrogen bonds with both S276 and D233 (Supplementary Figure 1f). These structural alterations may account for the differential residual activities observed for the K202R and K202Q mutants, approximately 58% and 12%, respectively (Supplementary Figure 1e). Overall, our findings suggest that mutations or acetylation of K202 may impair HDAC8 activity by disrupting the formation of the hydrogen bond network.

Decroos et al. 2015
(Left) **(Right)**

Figure for referee with unpublished data has been removed upon request by the authors.

New Supplementary Figure 1f

Supplementary Figure 1e

Figures for referee with unpublished data have been removed upon request by the authors.

Furthermore, we conducted molecular docking assays between HDAC8 and its substrate, SMC3 (Supplementary Figure 1g). Consistent with existing knowledge, we observed that the hydrophobic pocket of HDAC8 (WT) was occupied by the acetylated lysines (K105 and K106) of the substrate SMC3 during the deacetylation reaction. Notably, when docking mutants of HDAC8 to SMC3 in a similar manner, we discovered that both the K202R and K202Q mutations interfered with the accessibility of SMC3's acetylated lysines to the pocket. Intriguingly, the interference caused by the K202Q mutation was more pronounced. To validate these findings, we overexpressed Flag-HDAC8 or its mutants in 293T cells and subsequently performed immunoprecipitation. The results revealed that the HDAC8 K202R/K202Q mutants exhibited progressively weaker binding to the substrate SMC3 (Supplementary Figure 1h). These outcomes imply that K202R or K202Q mutations of HDAC8 may affect the structural stability by disrupting the hydrogen bond network, resulting in a weaker binding to substrates and thus a lower catalytic activity.

New Supplementary Figure 1g

Figure for referee with unpublished data has been removed upon request by the authors.

New Supplementary Figure 1h

Figure for referee with unpublished data has been removed upon request by the authors.

- It is unclear to me how the authors assessed the total amount of acetylation on HDAC8 in figure 2d. Did the authors first enrich for HDAC8 with a pulldown? Or does that this antibody recognize multiple acetylated residues on HDAC8?

Response: Thank you for your inquiry. In our experimental setup, we conducted Tip60 knockdown in cells overexpressing HDAC8 tagged with FLAG. Subsequently, we utilized anti-FLAG beads to immunoprecipitate HDAC8 and performed corresponding detection (Figure 2d). We have revised the respective figure accordingly.

New Figure 2d

Figure for referee with unpublished data has been removed upon request by the authors.

-It would be informative if the authors include Hos1, the *S. cerevisiae* cohesin deacetylase, in their evolutionary conservation alignments.

Response: Thank you and we have added the Hos1 into our evolutionary alignments. Noting the change in the corresponding residue sequence of Hos1 in *S. cerevisiae* to leucine instead of lysine is an important clarification.

Figure for referee with unpublished data has been removed upon request by the authors.

- The authors should include loading controls for all their western blot analyses.
- The authors should always show both the input and the pulldown for the IP experiments they perform. E.g. in figure 1g they only show the pulldown, but not the levels of AcK202, AcK, and FLAG in the input. In figure 2a and 2b, please show both the signal for FLAG and HA in the input.

Response: We thank the reviewer's suggestion. Following your recommendation, we have revised the corresponding figure and included the corresponding input data.

Figure for referee with unpublished data has been removed upon request by the authors.

Referee #3: This is a paper that attempts to find the acetylation of HDAC8 and investigate its physiological function. Hdac8 is a cohesin deacetylase that is important for regulation of cohesin dynamics, and thus regulates cohesin function in cell cycle and transcription. I think this paper has an impact, but I am concerned about many points as follows.

1) For example, in Fig 1f, they used a system in which HDAC8 expressed in E. coli is acetylated in E. coli, and it shows that the activity is reduced by K202ac. It certainly shows acetylation in K202, but acetylation of other residues must also be occurring. To prove that acetylation of K202ac reduces the activity of HDAC8, it is necessary to perform the same experiment with K202R.

Response: We appreciate the reviewer's concern, and we would like to provide an overview of the site-specific incorporation assay for inducing the acetylated modification in Figure 1f. This methodology was initially introduced by Neumann et al. in 2018 in Nature Chemical Biology (PMID: 18278036, cited by 702). To create a homogeneously K202-acetylated HDAC8 construct, we utilized a three-plasmid system (TEV-8, pCDFpyIT-1, and pAcKRS), detailed below. Wild-type HDAC8 was cloned into pTEV-8, yielding a C-terminal His6-tagged construct, and an amber codon was introduced at lysine 202 (AAG to TAG through site-directed mutagenesis). The amber construct was overexpressed in LB with spectinomycin (50 mg/ml), kanamycin (50 mg/ml), and ampicillin (150 mg/ml), along with 2mM N-acetyl-lysine and 20 mM nicotinamide to inhibit E. coli deacetylase activity during induction. The procedures for cell culture, expression, and purification were consistent with those outlined for recombinant human HDAC8. This system selectively yields total acetylated-K202 in HDAC8, preventing acetylation at other lysines. It has been successfully employed in purifying two distinct site-specific acetylated proteins in our prior research (Wei et al. 2018, PMID: 30755608, and Wan et al. 2020, PMID: 32783943). Additionally, we introduced the K202R mutation, mimicking the deacetylation status in HDAC8, to investigate its activity. The K202R variant exhibited a significantly reduced activity compared to wildtype HDAC8 (Supplemental Figure 1e), highlighting the critical role of K202 in HDAC8 activity and suggesting that acetylation at K202 may impact its deacetylase activity.

Figure 1f New Supplementary Figure 1e

Figures for referee with unpublished data have been removed upon request by the authors.

2) Fig4: They have data showing that the interaction in the TAD is stronger in cells heterozygous for K202Q (acetyl mimic) and that the loop gets also stronger. This mutation reduces the activity of HDAC8 (Fig1c). This means that the cohesin pool for loop extrusion is reduced. As far as I understand from other papers, this is strange. To resolve this discrepancy, I would like authors to do a basic check of the cells in which K202Q was introduced into the genome: what is the abundance of cohesin, what is the abundance in the chromatin fraction, what is the abundance of Smc3ac, etc.

Response: We acknowledge the reviewer's concern and have thoroughly verified our K202Q-KI cells through genome sequencing, confirming their accurate representation (Figure 4a). In our investigation, we illustrated that acetylated HDAC8 induces hyperacetylated SMC3, leading to enhanced chromatin loop formation. Recent work by Nagasaka et al. (2023, Molecular Cell, PMID: 37591243) supports our findings, revealing that treating HeLa cells with an HDAC8 inhibitor (PCI) increases SMC3 acetylation, resulting in a notable enhancement of chromatin loop formation. Moreover, a study by Wutz et al. (2022, eLife, PMID: 32065581) provides additional support, indicating that SMC3 acetylation can stabilize CTCF-anchored loops. In response to the reviewer's suggestion, we performed a basic immunoblotting analysis of the knock-in cells, presented in the new Figure 4b. This analysis revealed a significant elevation in SMC3 acetylation levels in the K202Q cells, while the HDAC8 expression level remained essentially unchanged compared to WT cells. These results further strengthen our assertion that acetylated HDAC8 plays a crucial role in modulating chromatin loop dynamics.

Figure 4a

New Figure 4b

Figures for referee with unpublished data have been removed upon request by the authors.

3) Fig5, d&e; the K202Q mutant cell line has a cohesion defect of over 70%. I wonder if such a high level of cohesion defects can cause death. I think that authors should perform a cohesion assay with a control such as siRad21 in parallel to see if there are any qualitative or quantitative differences.

Response: We appreciate the valuable suggestion from the reviewer. In response to your guidance, we conducted a chromosome spread assay with siRad21 as a positive control,

clearly demonstrating the knockdown effect of Rad21 in the new Supplementary Figure 5b. To provide a comprehensive understanding of the observed cohesion phenotypes, we categorized them into four distinct levels: normal, mild, moderate, and severe, as illustrated in the new Figure 5e. Statistical analysis revealed a significant increase in the proportion of cohesion defects in K202Q mutant cells, reaching approximately 67%. Notably, these defects were predominantly mild and moderate, with severe cohesion defects primarily observed in cells subjected to siRad21 treatment, in the new Figure 5d. This discrepancy can be attributed to the direct involvement of Rad21, a core subunit of the cohesion complex, in sister chromatid cohesion. In contrast, HDAC8 indirectly influences cohesion through the deacetylation modification of SMC3, another core subunit of cohesin. Additionally, we observed significant changes in the proliferation rates of different cell populations, as shown in the new Supplementary Figure 5c. As a positive control, cells treated with siRad21 exhibited pronounced growth inhibition, with the inhibitory effects appearing more prominent within 48 hours, aligning with the characteristics of transient transfection. In comparison to WT cells, the proliferation of K202R cells showed a modest decrease, while the proliferation inhibition observed in K202Q cells was more pronounced. These findings contribute to a more comprehensive understanding of the impact of HDAC8 mutations on cell cohesion and proliferation dynamics.

New Supplementary Figure 5b

New Figure 5e

Figures for referee with unpublished data have been removed upon request by the authors.

New Figure 5d

New Supplementary Figure 5c

Figures for referee with unpublished data have been removed upon request by the authors.

I raised here the most important points I feel that should be addressed at least. Overall the data contains lots of roughness. Therefore I could not recommend this paper to be published in any journal.

Response: In addition to the aforementioned points, it is imperative to further elaborate on

the enhancements across various aspects in the revised version. Primarily, we explicated the potential molecular mechanisms underlying the impact of the K202 mutation on enzyme activity. Drawing from Decroos et al.'s seminal work (2015, Biochemistry, PMID: 26463496), it is established that K202 plays a pivotal role in establishing a crucial hydrogen bond network (D233-K202-S276) essential for HDAC8 activity. Specifically, in wild-type (WT) HDAC8, the side chain of K202 orchestrates hydrogen bonds in concert with S276 and D233. Perturbation of this hydrogen bond network may precipitate diminished protein thermostability and enzymatic activity. In our investigation, we observed that the K202R mutation disrupts the hydrogen bond with S276 and attenuates the bond with D233, whereas the K202Q mutation results in the concurrent loss of hydrogen bonds with both S276 and D233 (Supplementary figure 1f). These structural modifications may elucidate the differential residual activities observed for the K202R and K202Q mutants, approximately 58% and 12%, respectively (Supplementary figure 1e). In summary, our findings underscore that mutations or acetylation of K202 may compromise HDAC8 activity by disrupting the formation of the hydrogen bond network.

Furthermore, we conducted molecular docking assays between HDAC8 and its substrate, SMC3 (Supplementary Figure 1g). Consistent with existing knowledge, we observed that the hydrophobic pocket of HDAC8 (WT) was occupied by the acetylated lysines (K105 and K106) of the substrate SMC3 during the deacetylation reaction. Notably, when docking mutants of HDAC8 to SMC3 in a similar manner, we discovered that both the K202R and K202Q mutations interfered with the accessibility of SMC3's acetylated lysines to the pocket. Intriguingly, the interference caused by the K202Q mutation was more pronounced. To validate these findings, we overexpressed Flag-HDAC8 or its mutants in 293T cells and subsequently performed immunoprecipitation. The results revealed that the HDAC8 K202R/K202Q mutants exhibited progressively weaker binding to the substrate SMC3 (Supplementary Figure 1h). These outcomes imply that K202R or K202Q mutations of HDAC8 may affect the structural stability by disrupting the hydrogen bond network, resulting in a weaker binding to substrates and thus a lower catalytic activity.

New Supplementary Figure 1f

Supplementary Figure 1e

Figures for referee with unpublished data have been removed upon request by the authors.

New Supplementary Figure 1g

Figure for referee with unpublished data has been removed upon request by the authors.

New Supplementary Figure 1h

Figure for referee with unpublished data has been removed upon request by the authors.

Additionally, we performed a replicative Hi-C experiment utilizing an additional site mutant clone. To address potential biases stemming from sequencing depth, we standardized valid contacts to maintain consistency across the three samples. Subsequent analysis yielded the outcomes depicted in the figure below. Notably, we have enhanced the resolution of the Hi-C maps in Figure 4i. Moreover, in a concerted effort to enhance transparency and facilitate thorough scrutiny, we have deposited the high-throughput sequencing data from both independent Hi-C experiments into the GEO database. This proactive measure is intended to provide unrestricted access to the raw data, fostering reproducibility and enabling fellow researchers to scrutinize and corroborate our findings.

New Figure 4d-f: Replicate 1 (normalized to **63 million** valid contacts per sample):

Figure for referee with unpublished data has been removed upon request by the authors.

New Supplementary Figure 4c-e: Replicate 2 (normalized to **56 million** valid contacts per sample):

Figure for referee with unpublished data has been removed upon request by the authors.

Dear Wei,

Thank you for submitting your revised manuscript. It has now been seen by all of the original referees.

As you can see, the referees find that the study is significantly improved during revision and recommend publication. However, I need you to address the points below before I can accept the manuscript.

- Please address the remaining concerns of the referees #2 and #3. Please contact me if you would like to discuss any of these further.
- Please provide 3-5 keywords for your study. These will be visible in the html version of the paper and on PubMed and will help increase the discoverability of your work.
- Please rename the Declarations of Interests section as "Disclosure Statement and Competing Interests".
- Please remove the Author Contributions section from the manuscript.
- As per our format requirements, in the reference list, citations should be listed in alphabetical order and then chronologically, with the authors' surnames and initials inverted; where there are more than 10 authors on a paper, 10 will be listed, followed by 'et al.'. Please see <https://www.embopress.org/page/journal/14693178/authorguide#referencesformat>
- Please fill out and include an author checklist as listed in our online guidelines (<https://www.embopress.org/page/journal/14693178/authorguide>)
- Main figures and Expanded View figures need to be submitted separately as one file per figure.
- Supplementary Figures need to be renamed as Expanded View figures and their callouts in the manuscript text need to be updated accordingly (e.g. Figure EV1, etc).
- We note the phrase 'data not shown' on page 3, which is not allowed as per journal policy. Please either show the data or remove the statements.
- The dataset GSE247611 needs to be made publicly accessible and the reviewer token needs to be removed from the Data Availability section and Source Data checklist. Moreover, a link that directly resolves to the dataset needs to be added to the Data Availability section.
- We note the following regarding the Source Data:
 - o Please resubmit source data as one zip file per figure.
 - o Source Data for Fig 1E: AcK202 blot is cropped to the same extent as the figure panel itself.
 - o Source Data for Fig 2B: blots western IP-HA HA and western IP-HA HDAC8 INPUT are too small.
- The manuscript sections should be in the following order: Title page - Abstract & Keywords - Introduction - Results - Discussion - Methods - Data Availability - Acknowledgments - Disclosure Statement & Competing Interests - References - Figure Legends - Tables with legends - Expanded View Figure Legends.
- Our production/data editors have asked you to clarify several points in the figure legends:
 - o Please define the annotated p values ****/**/* in the legend of figure 3b, d, f; 5f, supplementary figures 2c; 3e; 5c; as appropriate.
 - o Please indicate the statistical test used for data analysis in the legends of figures 1c, f; 2e; 3b-d, f; 4g-h; 5c, f, supplementary figures 1e; 2c; 3e; 5a-c.
 - o Please note that information related to n is missing in the legends of figures 3b-d, f; 4j; 5f, supplementary figures 2c; 3e.
 - o Please note that the error bars are not defined in the legends of figures 3b-d, f; 4j; 5f, supplementary figures 2c; 3e.
 - o Please note that scale bar and its definition are missing for figure 5e.
- Papers published in EMBO Reports include a 'synopsis' and 'bullet points' to further enhance discoverability. Both are displayed on the html version of the paper and are freely accessible to all readers. The synopsis includes a short standfirst summarizing the study in 1 or 2 sentences (max 35 words) that summarize the paper and are provided by the authors and streamlined by the handling editor. I would therefore ask you to include your synopsis blurb and 3-5 bullet points listing the key experimental findings.
- In addition, please provide an image for the synopsis. This image should provide a rapid overview of the question addressed in the study but still needs to be kept fairly modest since the image size cannot exceed 550 (width) x 300-600 (height) pixels.

Thank you again for giving us to consider your manuscript for EMBO Reports, I look forward to your minor revision.

Kind regards,

Deniz

--

Deniz Senyilmaz Tiebe, PhD
Editor
EMBO Reports

Referee #1:

The authors answered my concerns with appropriate experiments, and I can say that the paper is good enough for publication in *Embo Reports*.

Referee #2:

I would like to thank the authors for their efforts to improve the manuscript. However, there is one point that is still not clear. It concerns figure 3D (now new figure 3e): in this figure, we can see that synchronisation of wild-type cells allows us to appreciate the acetylation/deacetylation cycle of Smc3. I still think it would be important to assess this Smc3 acetylation/deacetylation cycle in cells expressing K202Q/R mutants by Smc3 immunoprecipitation (as they did for the new Figure 4B).

Referee #3:

We have read the revised manuscript by Sang et al. about a novel acetylation site on HDAC8 that controls HDAC8's deacetylase activity. In general we are happy with the changes that have been made to the manuscript. However, I have a strong concern regarding the HDAC8 K202R mutant cell lines.

The initial version of the manuscript showed Hi-C analyses on HDAC8 K202R cells, which displayed a subtle increase in the strength of primary loops. The requested replicate analyses on an independent clone showed that this effect was not reproducible. I presume that the authors therefore decided to not describe the Hi-C analyses for the HDAC8 K202R cell line in the text anymore. However, they still left the analyses on the primary clone in the main figures, including the correlation to changes in gene expression. In my opinion this is misleading, as these changes in gene expression are thus presumably not due to changes in 3D genome organization. It is also surprising to see that the HDAC8 K202R and HDAC8 K202Q cell lines display such different phenotypes in Hi-C analyses (Figure 4D and 4F), while the levels of cohesin acetylation appear similar between these cell lines (Figure 4B). The authors do not discuss this point. Taken together, the data displayed in Figure 4 and supplemental Figure 4 does not support the rather bold title of Figure 4, and ignore part of their data.

I also have some remaining textual comments. The authors describe the structures presented in supplemental Figure 1F as actual structures, and discuss how mutation affects the H-bonds in the catalytic site. However, I am missing information on how these structures are obtained in the methods. Are these new crystal structures, or are these predictions of how mutation of K202 in the published HDAC8 crystal structure (1W22) might affect the H-bonds? The authors should describe this more accurately in the text and methods. If these are indeed predictions, they should also rewrite their conclusions about these mutant forms of HDAC8. Similarly, the authors describe in supplemental Figure 1G that they provide a crystal structure of the SMC3-HDAC8 interaction, while this actually is a predicted structure that uses the crystal structure of HDAC8 and cryo-EM structure of SMC3. They don't describe how they obtained the HDAC8 mutant structures, which should be included in the methods.

Dear editors and reviewers,

We would like to thank the editors and reviewers for the consideration of our manuscript for publication in *EMBO Reports*. We have thus resubmitted the revised manuscript with a rebuttal letter that describes the changes point-by-point. Essentially, our report presents a comprehensive story about how the acetylated HDAC8 regulates the cell cycle, which will attract more interest to the fields of deacetylases, 3D genome and cell cycle.

As you can see, the referees find that the study is significantly improved during revision and recommend publication. However, I need you to address the points below before I can accept the manuscript.

Please find our point-to-point response below.

- Please address the remaining concerns of the referees #2 and #3. Please contact me if you would like to discuss any of these further.

Response: We have addressed the relevant concerns and the details have been sent to you earlier.

- Please provide 3-5 keywords for your study. These will be visible in the html version of the paper and on PubMed and will help increase the discoverability of your work.

Response: We have included keywords in the revised manuscript.

- Please rename the Declarations of Interests section as "Disclosure Statement and Competing Interests".

Response: We have already done so in the revised manuscript, as you requested.

665 ~~Disclosure Statement and Competing Interests~~↵

666 The authors declare no competing interests.↵

- Please remove the Author Contributions section from the manuscript.

Response: We have already done so in the revised manuscript, as you requested.

- As per our format requirements, in the reference list, citations should be listed in alphabetical order and then chronologically, with the authors'

surnames and initials inverted; where there are more than 10 authors on a paper, 10 will be listed, followed by 'et al.'. Please see <https://www.embopress.org/page/journal/14693178/authorguide#referencesformat>

Response: We have already done so in the revised manuscript, as you requested.

668 **References:**[↵]

- 669 Balasubramanian S, Ramos J, Luo W, Sirisawad M, Verner E, Buggy J (2008) A novel histone deacetylase
670 8 (HDAC8)-specific inhibitor PCI-34051 induces apoptosis in T-cell lymphomas. *Leukemia* 22: 1026-
671 1034[↵]
- 672 Beckouët F, Hu B, Roig MB, Sutani T, Komata M, Uluocak P, Katis VL, Shirahige K, Nasmyth K (2010)
673 An Smc3 acetylation cycle is essential for establishment of sister chromatid cohesion. *Mol Cell* 39: 689-
674 699[↵]
- 675 Beltrao P, Albanèse V, Kenner LR, Swaney DL, Burlingame A, Villén J, Lim WA, Fraser JS, Frydman J,
676 Krogan NJ (2012) Systematic functional prioritization of protein posttranslational modifications. *Cell*
677 150: 413-425[↵]
- 678 Ben-Shahar TR, Heeger S, Lehane C, East P, Flynn H, Skehel M, Uhlmann F (2008) Eco1-dependent
679 cohesin acetylation during establishment of sister chromatid cohesion. *Science* 321: 563-566[↵]

• Please fill out and include an author checklist as listed in our online guidelines (<https://www.embopress.org/page/journal/14693178/authorguide>)

Response: We have provided the author checklist, as you requested.

• Main figures and Expanded View figures need to be submitted separately as one file per figure.

Response: We have already done so in the revised manuscript, as you requested.

• Supplementary Figures need to be renamed as Expanded View figures and their callouts in the manuscript text need to be updated accordingly (e.g. Figure EV1, etc).

Response: We have already done so in the revised manuscript, as you requested.

• We note the phrase 'data not shown' on page 3, which is not allowed as per journal policy. Please either show the data or.

Response: We have already removed the statements in the revised manuscript. Thank you!

- The dataset GSE247611 needs to be made publicly accessible and the reviewer token needs to be removed from the Data Availability section and Source Data checklist. Moreover, a link that directly resolves to the dataset needs to be added to the Data Availability section.

Response: We have already done so, as you requested.

- We note the following regarding the Source Data:
 - o Please resubmit source data as one zip file per figure.
 - o Source Data for Fig 1E: Ack202 blot is cropped to the same extent as the figure panel itself.
 - o Source Data for Fig 2B: blots western IP-HA HA and western IP-HA HDAC8 INPUT are too small.

Response: We have already updated relevant data for Fig 1E and Fig 2B in the Source Data.

- The manuscript sections should be in the following order: Title page - Abstract & Keywords - Introduction - Results - Discussion - Methods - Data Availability - Acknowledgments - Disclosure Statement & Competing Interests - References - Figure Legends - Tables with legends - Expanded View Figure Legends.

Response: We have already done so in the revised manuscript, as you requested.

- Our production/data editors have asked you to clarify several points in the figure legends:
 - o Please define the annotated p values ****/**/* in the legend of figure 3b, d, f; 5f, supplementary figures 2c; 3e; 5c; as appropriate.
 - o Please indicate the statistical test used for data analysis in the legends of figures 1c, f; 2e; 3b-d, f; 4g-h; 5c, f, supplementary figures 1e; 2c; 3e; 5a-c.
 - o Please note that information related to n is missing in the legends of figures 3b-d, f; 4j; 5f, supplementary figures 2c; 3e.
 - o Please note that the error bars are not defined in the legends of figures 3b-d, f; 4j; 5f, supplementary figures 2c; 3e.

Response: We have carefully checked and revised the mentioned details.

- o Please note that scale bar and its definition are missing for figure 5e.

Response: We appreciate the reviewer's valuable suggestion and have added the scale bar into the Figure 5e.

Figure for referee with unpublished data has been removed upon request by the authors.

- Papers published in EMBO Reports include a 'synopsis' and 'bullet points' to further enhance discoverability. Both are displayed on the html version of the paper and are freely accessible to all readers. The synopsis includes a short standfirst summarizing the study in 1 or 2 sentences (max 35 words) that summarize the paper and are provided by the authors and streamlined by the handling editor. I would therefore ask you to include your synopsis blurb and 3-5 bullet points listing the key experimental findings.
- In addition, please provide an image for the synopsis. This image should provide a rapid overview of the question addressed in the study but still needs to be kept fairly modest since the image size cannot exceed 550 (width) x 300-600 (height) pixels.

Response: We have provided the above mentioned.

Referee #2:

I would like to thank the authors for their efforts to improve the manuscript. However, there is one point that is still not clear. It concerns figure 3D (now new figure 3e): in this figure, we can see that synchronisation of wild-type cells allows us to appreciate the acetylation/deacetylation cycle of Smc3. I still think it would be important to assess this Smc3 acetylation/deacetylation cycle in cells expressing K202Q/R mutants by Smc3 immunoprecipitation (as they did for the new Figure 4B).

Response: We appreciate the reviewer's valuable suggestion for improving the quality of our manuscript. We attempted to synchronize the K202Q/R cells with wild-type (WT) cells in the early S-phase using double thymidine arrest but were unsuccessful. Similar to our previous efforts to synchronize cells to the G1 phase using thymidine arrest, we successfully synchronized

WT HeLa cells (Results of attempted synchronization to G1 phase, left panel). However, synchronizing the mutant cells, particularly the K202Q mutants, proved challenging due to the varied proliferation statuses among these cells (Results of attempted synchronization to G1 phase, right panel), proved challenging due to varied proliferation statuses among mutant cells (Supplementary Figure 5c). This difficulty arises because K202Q/R cells exhibit significantly more cohesion defects compared to WT cells (Figure 5d-5e). Consequently, obtaining strictly synchronized K202R/Q mutant cells appears impractical, which makes conducting this experiment challenging.

Results of attempted synchronization to G1 phase Supplementary Figure 5c

Figure for referee with unpublished data has been removed upon request by the authors.

Figure 5d

Figure 5e

Figures for referee with unpublished data have been removed upon request by the authors.

Referee #3:

We have read the revised manuscript by Sang et al. about a novel acetylation site on HDAC8 that controls HDAC8's deacetylase activity. In general we are happy with the changes that have been made to the manuscript. However, I have a strong concern regarding the HDAC8 K202R mutant cell lines.

The initial version of the manuscript showed Hi-C analyses on HDAC8 K202R cells, which displayed a subtle increase in the strength of primary loops. The requested replicate analyses on an independent clone showed that this effect was not reproducible. I presume that the authors therefore decided to not describe the Hi-C analyses for the HDAC8 K202R cell line in the text anymore. However, they still left the analyses on the primary clone in the main figures, including the correlation to changes in gene expression. In my opinion this is misleading, as these changes in gene expression are thus presumably not due to changes in 3D genome organization. It is also surprising to see that the HDAC8 K202R and HDAC8 K202Q cell lines display such different phenotypes in Hi-C analyses (Figure 4D and 4F), while the levels of cohesin acetylation appear similar between these cell lines (Figure 4B). The authors do not discuss this point. Taken together, the data displayed in Figure 4 and supplemental Figure 4 does not support the rather bold title of Figure 4, and ignore part of their data.

Response: We appreciate the reviewer's valuable suggestions. In response to the concerns raised, we have quantified the results presented in Figure 4f and Supplementary Figure 4e using the GENOVA package in the latest revised manuscript. Specifically, we calculated the APA score to represent the average strength of the summed loops (New Figure 4f and New Supplementary Figure 4e) and conducted a statistical analysis of the fold change of loops in the APA analysis (Results of foldchange of loops).

New Figure 4f (Replicate 1)

New Supplementary Figure 4e (Replicate 2)

Figures for referee with unpublished data have been removed upon request by the authors.

Results of foldchange of loops (Replicate 1(left), Replicate 2 (right))

Figure for referee with unpublished data has been removed upon request by the authors.

As the reviewer noted, our initial submission showed a slight increase in chromatid loop strength in K202R mutant cells compared to WT cells (New Figure 4f, WT: 17.3 vs. K202R: 18.7; Results of foldchange of loops, Replicate 1). However, this difference was negligible in our second Hi-C replicate (New Figure 4f, WT: 22.8 vs. K202R: 22.5; Results of foldchange of loops, Replicate 2). Hi-C experiments inherently introduce unavoidable noise, making small differences hard to detect consistently across samples. To maintain rigor, we have revised our manuscript to omit the claim of increased chromatin loop strength in K202R cells. However, this does not imply that the K202R mutation has no impact on 3D chromatin structure. On the contrary, both Hi-C replicates (Figure 4d and Supplementary Figure 4c) demonstrate that the K202R mutation alters chromatin interactions, indicating changes in the 3D structure of chromatin. These changes may not be apparent at the level of TADs and loops, but as our understanding of chromatin architecture evolves, we may better decipher these alterations. In summary, we believe it is valuable to retain the K202R data in Figures 4d, e, f.

Figure 4d

Supplementary Figure 4c

Figures for referee with unpublished data have been removed upon request by the authors.

Moreover, we considered moving Figure 4f to the supplementary figures (Supplementary Figure 4f), which would not impact our conclusions in Figure 4. The 3D genome structure, including chromatin loops, is dynamically regulated within cells. Our Hi-C data analysis revealed that while some chromatin loops in K202R mutant cells increased, others decreased compared to WT cells, resulting in only small differences observed in the APA analysis (New Figure 4f and New Supplementary Figure 4e). This

variability may explain the significant correlation between K202R and WT (Supplementary Figure 4f). Similarly, K202Q cells displayed generally enhanced chromatin loop strength, resulting in a stronger positive correlation (Figure 4g).

Supplementary Figure 4f

Figure 4g

Figures for referee with unpublished data have been removed upon request by the authors.

Finally, in response to the reviewer's concern about the new Figure 4b, we quantitatively analyzed the Western blot results, revealing a significant increase in SMC3 acetylation in K202Q mutant cells compared to K202R mutant cells (New Figure 4b). Analysis of two replicate Hi-C experiments (Figure 4d and Supplementary Figure 4c) showed distinct alterations in chromatin interactions between K202R and K202Q mutant cells compared to WT cells, suggesting a correlation with the differentially increased levels of SMC3 acetylation (New Figure 4b). These findings underscore the vital regulatory role of HDAC8-mediated SMC3 acetylation in chromatin architecture. Notably, the K202R mutation did not significantly affect TAD and loop formation overall, likely because these conserved structures could be sufficiently maintained by the residual HDAC8 activity. In contrast, the K202Q mutation, with lower residual HDAC8 activity (Supplementary Figure 1e), significantly disrupted chromatin structure, including TADs and loops, further highlighting the critical role of HDAC8 acetylation in regulating 3D genome architecture.

New Figure 4b

Supplementary Figure 1e

Figures for referee with unpublished data have been removed upon request by the authors.

We have added a description of the variability of Hi-C experiments of K202R mutant cells in the *Results* and *Discussion* sections of the revised manuscript, respectively.

Results

302 made in HeLa cells treated with the HDAC8 inhibitor PCI (Nagasaka *et al*, 2023).
303 RCP analysis of two independent Hi-C experiments revealed consistent
304 alterations in chromatin interactions in K202R mutant cells compared to WT
305 cells, though these alterations were less pronounced than those observed in
306 K202Q mutant cells. Notably, no evident changes were consistently observed
307 in the aggregate intensity of TADs and chromatin loops between K202R and
308 WT cells. Gene expression is intricately regulated, and the spatiotemporal

Discussion

438 which consequently favors the stable binding of cohesin to DNA, restricting its
439 sliding across chromatin. However, while consistent alterations in the frequency
440 of chromatin interactions were observed in K202R mutant cells, alterations in
441 the overall intensities of TADs and loops were not invariably apparent. This
442 observation indicates that although the K202R mutation does alter the 3D
443 structure of chromatin to some extent, its influence on the formation of TADs
444 and chromatin loops is relatively limited. One plausible explanation is that the
445 residual enzyme activity of the K202R mutation is largely sufficient to maintain
446 overall TAD and loop regulation. Consequently, potential subtle deviations
447 induced by the K202R mutation are challenging to detect reliably due to the
448 inherent experimental noise associated with Hi-C techniques.

I also have some remaining textual comments. The authors describe the structures presented in supplemental Figure 1F as actual structures, and discuss how mutation affects the H-bonds in the catalytic site. However, I am missing information on how these structures are obtained in the methods. Are these new crystal structures, or are these predictions of how mutation of K202 in the published HDAC8 crystal structure (1W22) might affect the H-bonds? The authors should describe this more accurately in the text and methods. If these are indeed predictions, they should also rewrite their conclusions about these mutant forms of HDAC8. Similarly, the authors describe in supplemental Figure 1G that they provide a crystal structure of

the SMC3-HDAC8 interaction, while this actually is a predicted structure that uses the crystal structure of HDAC8 and cryo-EM structure of SMC3. They don't describe how they obtained the HDAC8 mutant structures, which should be included in the methods.

Response: We thank the reviewer for this suggestion. As noted, the K202R and K202Q mutations were indeed predicted using the software PyMOL. Given that the crystal structure of HDAC8-WT has been determined and published (PDB accession code 1W22), utilizing the mutagenesis function of PyMOL based on the HDAC8-WT crystal structure to predict the structure of point mutations is a viable approach. Following the reviewer's suggestion, we have described this process more accurately in the manuscript and figure legends of the latest revised version to facilitate a better and more precise understanding of our study.

Latest revised version (Manuscript)

According to previous study by Decroos et al., K202 plays a crucial role in establishing a hydrogen bond network (D233-K202-S276) essential for HDAC8 activity.¹⁹ By utilizing PyMOL to predict structure of K202R/Q mutations, we observed that the K202R mutation disrupted the hydrogen bond with S276 and weakened the bond with D233, while the K202Q mutation led to the simultaneous loss of hydrogen bonds with both S276 and D233 (supplementary Figure 1f). These structural alterations may account for the differential residual

Latest revised version (Figure legends)

performed activity assay. Data are presented as mean \pm SD of three duplicate experiments. **** $p < 0.001$. f. HDAC8 K202Q mutation may disrupt the hydrogen bond network involving D233-K202-S276. The crystal structure of HDAC8-WT was obtained from previously published data (PDB accession code 1W22). The K202Q and K202R mutant structures of HDAC8 were predicted using the mutagenesis function of PyMOL based on the HDAC8-WT crystal structure. Dotted yellow lines indicated hydrogen bonds. In HDAC8-WT, robust

Dear Prof. Yu,

Before we can accept the manuscript, the following remaining points need to be addressed:

- Please add a discussion point into the text acknowledging the variability in the chromatid loop strength of K202R mutant cells (as pointed out by referee #3).
- Please add a scale bar to Figure 5e and define its length in the figure legends.
- Please specify the nature of the replicates stated in the figure legends (i.e. biological, technical).

Many thanks.

Your paper has been placed back in the Author Approval Folder where you may access via the following link:

Link Unavailable

Please make the correction(s) as specified above and resubmit your paper following the same steps as before.

Should you have any queries, please do not hesitate to contact us.

Kind regards,
Bojana

Bojana Perkucin
Editorial Assistant
EMBO Press

- Please add a discussion point into the text acknowledging the variability in the chromatid loop strength of K202R mutant cells (as pointed out by referee #3).

Response: We have added a description of the variability of Hi-C experiments of K202R mutant cells in the *Results* and *Discussion* sections of the revised manuscript, respectively.

Results

302 made in HeLa cells treated with the HDAC8 inhibitor PCI (Nagasaka *et al*, 2023).
303 RCP analysis of two independent Hi-C experiments revealed consistent
304 alterations in chromatin interactions in K202R mutant cells compared to WT
305 cells, though these alterations were less pronounced than those observed in
306 K202Q mutant cells. Notably, no evident changes were consistently observed
307 in the aggregate intensity of TADs and chromatin loops between K202R and
308 WT cells. Gene expression is intricately regulated, and the spatiotemporal

Discussion

438 which consequently favors the stable binding of cohesin to DNA, restricting its
439 sliding across chromatin. However, while consistent alterations in the frequency
440 of chromatin interactions were observed in K202R mutant cells, alterations in
441 the overall intensities of TADs and loops were not invariably apparent. This
442 observation indicates that although the K202R mutation does alter the 3D
443 structure of chromatin to some extent, its influence on the formation of TADs
444 and chromatin loops is relatively limited. One plausible explanation is that the
445 residual enzyme activity of the K202R mutation is largely sufficient to maintain
446 overall TAD and loop regulation. Consequently, potential subtle deviations
447 induced by the K202R mutation are challenging to detect reliably due to the
448 inherent experimental noise associated with Hi-C techniques.

- Please add a scale bar to Figure 5e and define its length in the figure legends.

Response: We have already done so in the revised manuscript, as you requested.

Figure for referee with unpublished data has been removed upon request by the authors.

- Please specify the nature of the replicates stated in the figure legends (i.e. biological, technical).

Response: We have already done so in the revised manuscript, as you requested.

Prof. Wei Yu
Fudan University
SONGHU ROAD 2005
BIOLOGICAL BUILDING D303
SHANGHAI 200438
China

Dear Prof. Yu,

Thank you for submitting your revised manuscript. I have now looked at everything and all is fine. Therefore, I am very pleased to accept your manuscript for publication in EMBO Reports.

Congratulations on a nice work!

Before we can transfer your manuscript to our production team, I need your input on one more point - I made some minor changes in the items below. Please take a look and confirm, or feel free to propose further changes. Thank you.

Title:

Reversible Acetylation of HDAC8 Regulates Cell Cycle

Abstract:

HDAC8, a member of class I HDACs, plays a pivotal role in cell cycle regulation by deacetylating the cohesin subunit SMC3. While cyclins and CDKs are well-established cell cycle regulators, our knowledge of other regulators remains limited. Here we reveal the acetylation of K202 in HDAC8 as a key cell cycle regulator responsive to stress. K202 acetylation in HDAC8, primarily catalyzed by Tip60, restricts HDAC8 activity, leading to increased SMC3 acetylation and cell cycle arrest. Furthermore, cells expressing the mutant form of HDAC8 mimicking K202 acetylation display significant alterations in gene expression, potentially linked to changes in 3D genome structure, including enhanced chromatid loop interactions. K202 acetylation impairs cell cycle progression by disrupting the expression of cell cycle-related genes and sister chromatid cohesion, resulting in G2/M phase arrest. These findings indicate the reversible acetylation of HDAC8 as a cell cycle regulator, expanding our understanding of stress-responsive cell cycle dynamics.

Kind regards,

Deniz Senyilmaz Tiebe

--

Deniz Senyilmaz Tiebe, PhD
Editor
EMBO Reports

--
